# Metabolic characteristics of CD8+ T cell subsets in young and aged individuals are not predictive of functionality

Kylie M. Quinn [1,2✉], Tabinda Hussain [1], Felix Kraus[1], Luke E. Formosa [1], Wai K. Lam [1], Michael J. Dagley [3], Eleanor C. Saunders[3], Lisa M. Assmus [1,4], Erica Wynne-Jones[5], Liyen Loh [5], Carolien E. van de Sandt [5,6], Lucy Cooper [1], Kim L. Good-Jacobson [1], Katherine Kedzierska [5], Laura K. Mackay [5], Malcolm J. McConville [3], Georg Ramm [3,7], Michael T. Ryan [1] & Nicole L. La Gruta [1✉]

Virtual memory T ($T_{VM}$) cells are antigen-naïve CD8+ T cells that exist in a semi-differentiated state and exhibit marked proliferative dysfunction in advanced age. High spare respiratory capacity (SRC) has been proposed as a defining metabolic characteristic of antigen-experienced memory T ($T_{MEM}$) cells, facilitating rapid functionality and survival. Given the semi-differentiated state of $T_{VM}$ cells and their altered functionality with age, here we investigate $T_{VM}$ cell metabolism and its association with longevity and functionality. Elevated SRC is a feature of $T_{VM}$, but not $T_{MEM}$, cells and it increases with age in both subsets. The elevated SRC observed in aged mouse $T_{VM}$ cells and human CD8+ T cells from older individuals is associated with a heightened sensitivity to IL-15. We conclude that elevated SRC is a feature of $T_{VM}$, but not $T_{MEM}$, cells, is driven by physiological levels of IL-15, and is not indicative of enhanced functionality in CD8+ T cells.

[1] Department of Biochemistry and Molecular Biology, Monash Biomedicine Discovery Institute, Monash University, Clayton, VIC 3800, Australia. [2] School of Health and Biomedical Sciences, RMIT University, Bundoora, VIC, Australia. [3] Department of Biochemistry and Molecular Biology, Bio21 Institute of Molecular Science and Biotechnology, University of Melbourne, Parkville, VIC 3010, Australia. [4] Institute of Experimental Immunology, University Hospital Bonn, 53127 Bonn, Germany. [5] Department of Microbiology and Immunology, University of Melbourne, Peter Doherty Institute for Infection and Immunity, Parkville, VIC, Australia. [6] Department of Hematopoiesis, Sanquin Research and Landsteiner Laboratory, Amsterdam UMC, University of Amsterdam, 1066CX Amsterdam, Netherlands. [7] Monash Ramaciotti Centre for Cryo-EM, Monash University, Clayton, VIC, Australia. ✉email: Kylie.quinn@rmit.edu.au; Nicole.la.gruta@monash.edu

Previous studies have highlighted that the metabolic phenotype of CD8[+] T cells dramatically impacts their functional and survival capacities (reviewed in refs. [1,2]). True naive ($T_N$) cells are quiescent and predominantly utilise oxidative phosphorylation (OXPHOS) to meet their low energy demands. In contrast, effector ($T_{EFF}$) cells undergo transcriptional reprogramming to upregulate aerobic glycolysis after TCR stimulation. Conventional memory ($T_{MEM}$) cells revert to the predominant utilisation of OXPHOS but are characterised by a higher mitochondrial energy reserve, known as spare respiratory capacity (SRC)[3–5]. SRC is the difference between basal and maximal oxygen consumption rates (OCR)[4,5] and it reflects the mitochondrial capacity that a cell holds in reserve, which may mitigate stress from sudden increases in energy demand. Increased SRC has been proposed to mediate both enhanced T cell functionality, in the form of metabolic memory that confers immediate responsiveness after secondary antigen exposure[5], and the increased longevity of $T_{MEM}$ cells[5,6].

The greater SRC observed in $T_{MEM}$ cells was in turn associated with increased mitochondrial load and, in particular, a distinct fused mitochondrial morphology when compared to $T_N$ and $T_{EFF}$ cells[5,6]. Mitochondrial fusion was causally linked with $T_{MEM}$ formation and function, since deletion of the inner mitochondrial membrane fusion protein, Opa1, abrogated the development of $T_{MEM}$ cells after infection, while promoting mitochondrial fusion in $T_{EFF}$ cells conferred a memory phenotype[6]. More recently, high SRC was shown to partition preferentially with a subset of long-lived $T_{MEM}$ cells known as central memory ($T_{CM}$) cells, rather than short-lived effector memory ($T_{EM}$) cells. Enforcing glycolysis, rather than OXPHOS, in CD8[+] $T_{MEM}$ cells limited their ability to survive and establish the long-lived $T_{CM}$ population[7]. Importantly, many studies on metabolic characteristics of $T_{MEM}$ cells have utilised memory phenotype cells generated in vitro in the context of high levels of IL-15 (refs. [3–6,8]), but the specific impact of IL-15 on these metabolic characteristics has not been well defined.

$T_{VM}$ cells are a subset of antigen-naive, semi-differentiated CD8[+] T cells. They are generated in neonatal mice[9,10] independently of antigen exposure, as evidenced by their presence in germ-free mice, antigen-free mice and CD8[+] T cell populations specific for viral antigens in naive mice[11–14]. Common γ (γc) chain cytokine signalling is thought to drive the semi-differentiated phenotype of $T_{VM}$ cells, likely via homoeostatic proliferation, with IL-15 transpresentation by CD8α[+] dendritic cells (DCs) required for their generation[11,15], and they appear to develop from T cells with modestly self-reactive TCRs[13,14,16–18]. Although antigenically naive, $T_{VM}$ cells are functionally distinct from true naive T ($T_N$) cells, as $T_{VM}$ engage proliferation and cytokine production more rapidly upon TCR stimulation and can also respond to cytokine stimulation[17,19]. $T_{VM}$ cells are also phenotypically distinct from both $T_N$ and $T_{MEM}$ cells, with high levels of CD44, a classical marker of activation, but low levels of CD49d, which is only upregulated upon strong cognate antigen encounter[11,20]. Of note, the high level expression of CD44 and CD62L on $T_{VM}$ cells has resulted in their frequent misclassification as central memory ($T_{CM}$) cells, if CD49d expression is not assessed[20].

While $T_{VM}$ cells exhibit augmented function in the young, ageing dramatically undermines the functionality of both $T_{VM}$ and $T_{MEM}$ subsets in vitro and in vivo[17,21,22]. Moreover, while $T_N$ cells decline substantially in number and proportion, both $T_{VM}$ and $T_{MEM}$ cells accumulate with advanced age[17,20,23,24], with the accumulation of $T_{VM}$ cells partially dependent on type I IFN-related signalling pathways[25]. Overall, it is unclear whether the metabolic profile of CD8[+] T cell subsets changes with age and whether metabolic changes reflect their capacity for function and survival over the lifespan.

Given that $T_{VM}$ cells span the phenotypic and functional divide between $T_N$ and $T_{MEM}$ cells, we aim to dissect the metabolic characteristics and associated mitochondrial features of this unique T cell subset to determine whether these features are indicative of their function and survival capacity during ageing. We define $T_{MEM}$ cells as only those that have encountered antigen, which necessitates a reanalysis of the metabolic, survival, and functional characteristics of bona fide $T_{MEM}$ cells. Collectively, this study refines our understanding of how the metabolic state impacts on CD8[+] T cell function and longevity, which is ultimately key to augmenting or suppressing T cell function using clinical interventions.

## Results

**$T_{VM}$ and cells from aged mice have increased spare respiratory capacity.** We sought to understand whether the basal metabolic phenotype of $T_{VM}$ cells is more closely aligned with the $T_N$ population or shares characteristics with conventional $T_{MEM}$ cells, such as increased mitochondrial load and SRC, and to understand how these profiles change with age. We, therefore, undertook a comprehensive mitochondrial and metabolic analysis for each of these subsets isolated from the spleens of naive young and aged specific-pathogen-free (SPF) mice.

The basal mitochondrial metabolic profile of young and aged mouse CD8[+] T cells subsets was determined by performing a Mito Stress test, using a Seahorse XFe96 Bioanalyser, on sorted $T_N$ (CD44[lo]), $T_{VM}$ (CD44[hi]CD49d[lo]) and $T_{MEM}$ (CD44[hi]CD49d[hi]) CD8[+] cells directly ex vivo. In this assay, oxygen consumption rate (OCR) is tracked in the basal state ($OCR_{Bas}$) and then during treatment with various mitochondrial inhibitors to enable measurement of maximal OCR ($OCR_{Max}$), with the difference between the $OCR_{Bas}$ and $OCR_{Max}$ representing SRC. $T_N$ cells from young and aged mice had comparable OCR profiles, resulting in comparable SRC for these subsets (Fig. 1a, b). $T_{VM}$ cells from young mice had significantly higher OCR than $T_N$ cells, at both $OCR_{Bas}$ and, more noticeably, at $OCR_{Max}$, resulting in significantly increased SRC (Fig. 1a, b), consistent with a memory-like phenotype. The $T_{VM}$ cells from aged mice also had consistently higher OCR than those from young mice leading to substantially higher SRC (Fig. 1a, b). Strikingly, $T_{MEM}$ cells from young mice did not exhibit higher SRC than $T_N$ cells, but rather had the lowest SRC of all subsets (Fig. 1a, b), in contrast to previous reports[4]. While a significant increase in SRC was also observed in $T_{MEM}$ cells with age (Fig. 1a, b), it remained significantly lower than the SRC observed for $T_{VM}$ cells from aged mice. These data suggest that high SRC is not a hallmark of $T_{MEM}$ cells but instead defines $T_{VM}$ cells, and that ageing drives an increase in SRC in all memory phenotype CD8[+] T cells.

**Correlation of SRC mitochondrial characteristics.** Increased SRC is often thought to reflect quantitative and/or qualitative changes in the mitochondria themselves, including (i) denser mitochondrial cristae, promoting processivity and efficiency of oxygen consumption by the electron transport chain (ETC)[3], (ii) a fused mitochondrial morphology[6] and (iii) increased mitochondrial load or volume[5]. To assess these mitochondrial characteristics, $T_N$, $T_{VM}$ and $T_{MEM}$ cells from young and aged mice were sorted and mitochondria were imaged directly ex vivo. There were no obvious differences in the size, density or morphology of mitochondrial cristae across cell types or ages, by electron microscopy (Fig. 1c). When mitochondrial fusion was scored by confocal microscopy (Fig. 1d), an intermediate or fused morphology was observed only in a minority of cells, even for $T_{MEM}$ cells (Fig. 1e), suggesting that mitochondrial fusion is not required for $T_{MEM}$ cell maintenance. In addition, no $T_{VM}$ cells

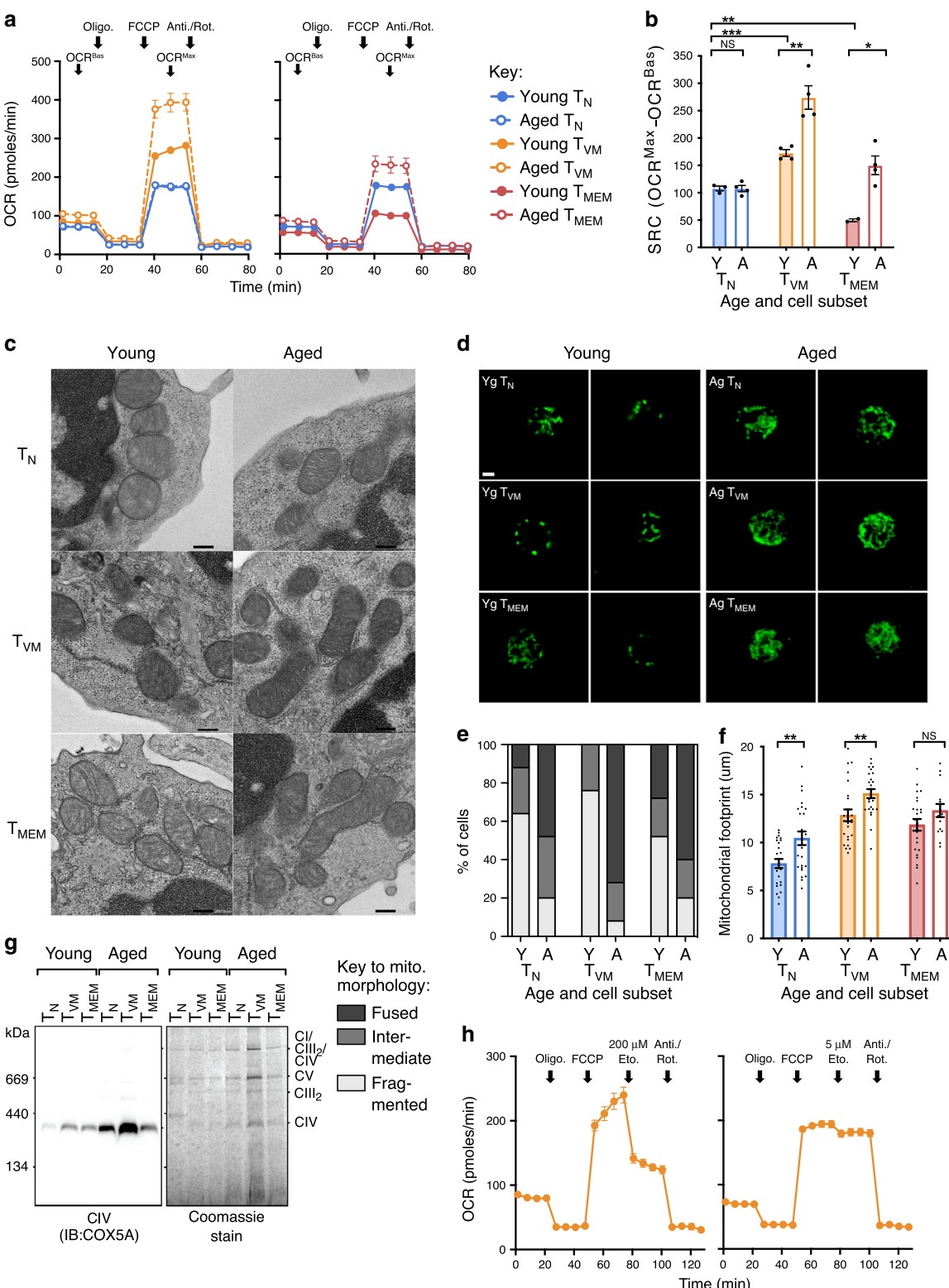

from young mice were observed with a fused morphology (Fig. 1e), despite their high SRC. Strikingly, there was a substantial increase in mitochondrial fusion with age across all cell types, with this effect being most apparent in $T_{VM}$ cells (Fig. 1e). Finally, mitochondrial footprint per cell as a measure of mitochondrial load was highest in $T_{VM}$ and $T_{MEM}$ cells and it increased significantly with age in $T_N$ cells and $T_{VM}$ cells (Fig. 1f).

Generally, mitochondrial cristae morphology, fusion or volume correlated poorly with the high SRC observed selectively in the $T_{VM}$ cell subset. To obtain a more direct measure of Electron Transport Chain (ETC) capacity, the levels of mitochondrial ETC Complex IV (CIV; Cox5a) from a defined number of each cell subset was quantitated directly via blue-native PAGE and immunoblotting. Although not an absolute correlation, the

**Fig. 1 $T_{VM}$ cells have high SRC and CIV, which increases with age. a** Oxygen consumption rate (OCR) across time for sorted $T_N$, $T_{VM}$ and $T_{MEM}$ cells from the spleens of naive young and aged SPF mice. Arrows indicate the addition of mitochondrial inhibitors (oligomycin; FCCP; antimycin A/rotenone) or timepoints for assessment of SRC (OCR$_{Bas}$, OCR$_{Max}$) ($n = 2$–$4$, 5 experimental replicates). **b** Change in OCR from OCR$_{Bas}$ to OCR$_{Max}$ (SRC) for each sorted subset ($n = 2$–$4$, 5 experimental replicates). **c** Electron microscope images of sorted cells directly ex vivo, scale bar indicates 0.2 μm (1 experimental replicate). **d** Confocal microscopy of sorted cells directly ex vivo, green fluorescence is Cytochrome C staining, scale bar indicates 2 μm, which was used to define (**e**), predominant mitochondrial morphology (fused, intermediate or fragmented) for 140 cells per subset and **f** average mitochondrial footprint per cell as calculated from confocal images (3 experimental replicates). **g** BN-PAGE and Blot for Cox5a from ETC CIV, with bands for CIV, CII, CIII$_2$, CV and the CI/CIII$_2$/CIV supercomplex indicated, alongside the Coomassie stained blot (3 experimental replicates). **h** Oxygen consumption rate (OCR) across time for sorted $T_{VM}$ cells from young mice, with high (200 μM) or low (5 μM) dose Etomoxir, ($n = 3$, 3 experimental replicates). Shown is mean ± standard error of the mean (SEM). NS indicates not significant, * indicates $p \leq 0.05$, ** indicates $p \leq 0.01$, unpaired $t$ test.

amount of CIV appeared to correlate better with SRC than mitochondrial load or morphology; namely CIV was increased in $T_{VM}$ compared to $T_N$ cells from young mice, and age-related increases in CIV were most marked in the $T_{VM}$ population (Fig. 1g). Collectively, our analyses of mitochondrial load and morphology suggested that they were broadly predictive of age-related increases in SRC. Expression levels of ETC CIV appeared to most accurately predict cellular SRC across age and subsets.

To provide a mechanistic basis for increased mitochondrial load/activity in $T_{VM}$ cells from aged mice, RNA-Seq data previously generated from $T_N$, $T_{VM}$, and $T_{MEM}$ subsets from young and aged mice[17] was interrogated for transcripts involved in mitochondrial biogenesis, mitophagy, and mitochondrial fusion or fission. Across all T cell subsets there was an age-dependent decrease in *Atg101* and *Ulk1* transcripts, which are critical for mitophagy (Supplementary Fig. 1a). There was a corresponding increase in PGC-1α transcripts (*Ppargcla*) with age in both memory phenotype populations, which was particularly striking in the $T_{VM}$ subset (Supplementary Fig. 1a). Analysis of transcript levels associated with mitochondrial fusion (*Mfn1*, *Mfn2*, *Opa1*) or fission (*Dnm1l*) revealed minimal to no change across T cell subsets or with aging (Supplementary Fig. 1c). Collectively, whilst true delineation of the impact of mitochondrial dynamics on mitochondrial load requires more detailed biochemical analyses, these transcriptional data highlight that there may be a decrease in mitochondrial degradation and an increase in biogenesis with age, particularly in the $T_{VM}$ subset, which may drive the observed increase in mitochondrial load and SRC.

**No evidence of FAO fuelling high SRC in $T_{VM}$ cells.** High SRC in $T_{MEM}$ cells was previously thought to be fuelled by fatty acid oxidation (FAO), a mechanism largely defined using etomoxir to inhibit carnitine palmitoyltransferase I (Cpt1), which is a rate-limiting enzyme for FAO[3]. However, it was recently demonstrated that the high concentration of etomoxir used in these studies also inhibited other components of OXPHOS to reduce SRC[26–28]. To examine the impact of etomoxir on high SRC in $T_{VM}$ cells, the drug was incorporated into the Mito Stress assay with young mouse $T_{VM}$ cells at either a high concentration (200 μM) or a low concentration (5 μM), the latter of which is predicted to maintain specificity for Cpt1 (ref.[27]). A substantial decrease in OCR$_{Max}$ was observed with the high concentration of etomoxir, but there was only a very modest decrease in OCR$_{Max}$ with the low concentration (Fig. 1h). This suggests that FAO, via Cpt1, does not facilitate the high SRC observed in $T_{VM}$ cells.

We next investigated the possibility that glycolysis was required to fuel the high SRC observed in $T_{VM}$ cells, most likely via the production of pyruvate[29]. The Mito Stress assay was performed on $T_{VM}$ cells from naive mice with the addition of 2-Deoxy-D-glucose (2-DG), a glucose analogue that inhibits glycolysis. The addition of 2-DG had a minimal effect on OCR (Supplementary Fig. 2a), similar to the addition of low dose etomoxir (Fig. 1h). By

contrast, 2-DG addition dramatically reduced ECAR, confirming its effective inhibition of glycolysis (Supplementary Fig. 2b). These data suggest that the high basal SRC observed in $T_{VM}$ cells is not exclusively dependent on either FAO or glycolysis, but may be fuelled by a substrate generated independently of both pathways.

**Virus infection drives increased SRC in $T_{VM}$ cells.** High SRC was not observed in $T_{MEM}$ cells in this study (Fig. 1a, b), but these $T_{MEM}$ cells were isolated out of naive SPF mice and were likely generated in response to commensal or low-pathogenicity organisms and in conditions of low inflammation. To assess SRC in infection-generated $T_{MEM}$ cells, mice were infected with influenza A virus (IAV) and $T_{MEM}$ cells specific for tetrameric H-$2D^b$ loaded with NP$_{366}$, PA$_{224}$ and PB1-F2$_{62}$ epitopes ($T_{MEM}$ (IAV) cells) were isolated 20 days later. Both $T_{MEM}$ and $T_{MEM}$ (IAV) cells exhibited a similarly low SRC (Fig. 2a, b). Interestingly, $T_{MEM}$ (IAV) cells consistently exhibited substantially higher basal and maximal extracellular acidification rates (ECAR$_{Bas}$ and ECAR$_{Max}$), compared to $T_N$ and $T_{MEM}$ cells (Fig. 2c). These data illustrate that $T_{MEM}$ and $T_{MEM}$ (IAV) cells are metabolically distinct with regard to glycolytic, but not OXPHOS, capacity.

Strikingly, recent IAV infection caused a substantial elevation in the SRC of $T_{VM}$ cells (Fig. 2d, e), without any shift in glycolytic capacity (Fig. 2f). These data demonstrate that infection, like ageing, leads to an environment that augments SRC selectively in $T_{VM}$ cells (and thus in an antigen-independent manner), and reinforce that high SRC is not a canonical feature of $T_{MEM}$ cells, even those induced by infection.

**Conventionally defined $T_{CM}$ cells are predominantly $T_{VM}$ Cells.** Recently, high SRC was shown to partition preferentially with the long-lived central memory ($T_{CM}$; CD44$^{hi}$CD62L$^{hi}$) subset of $T_{MEM}$, rather than short-lived effector memory ($T_{EM}$; CD44$^{hi}$CD62L$^{lo}$) cells[7]. In that study, $T_{CM}$ cells appear to have been defined as CD44$^{hi}$CD62L$^{hi}$ CD8$^+$ T cells obtained from mice after acute lymphocytic choriomeningitis virus (LCMV) infection, which would include $T_{VM}$ cells[20]. To determine the extent to which metabolic characteristics of $T_{CM}$ cells have been conflated with those of $T_{VM}$ cells, we assessed the proportion of classically defined $T_{CM}$ cells (CD44$^{hi}$CD62L$^{hi}$) that were actually $T_{VM}$ cells (CD44$^{hi}$CD62L$^{hi}$CD49d$^{lo}$) in naive young, naive aged or LCMV-infected mice. In young and aged naive mice, the vast majority ($\geq 85\%$) of $T_{CM}$ cells were found to be CD49d$^{lo}$ and therefore $T_{VM}$ cells (Fig. 3a). Even 40 days after acute LCMV infection, which induces a substantial CD8$^+$ T cell response and establishes robust antigen-specific memory populations[30], over 60% of $T_{CM}$ cells were found to be $T_{VM}$ cells (Fig. 3a). This highlights the possibility that CD8$^+$ T cell populations previously defined as $T_{CM}$ cells, from young, aged or infected mice, may have been predominantly comprised of $T_{VM}$ cells.

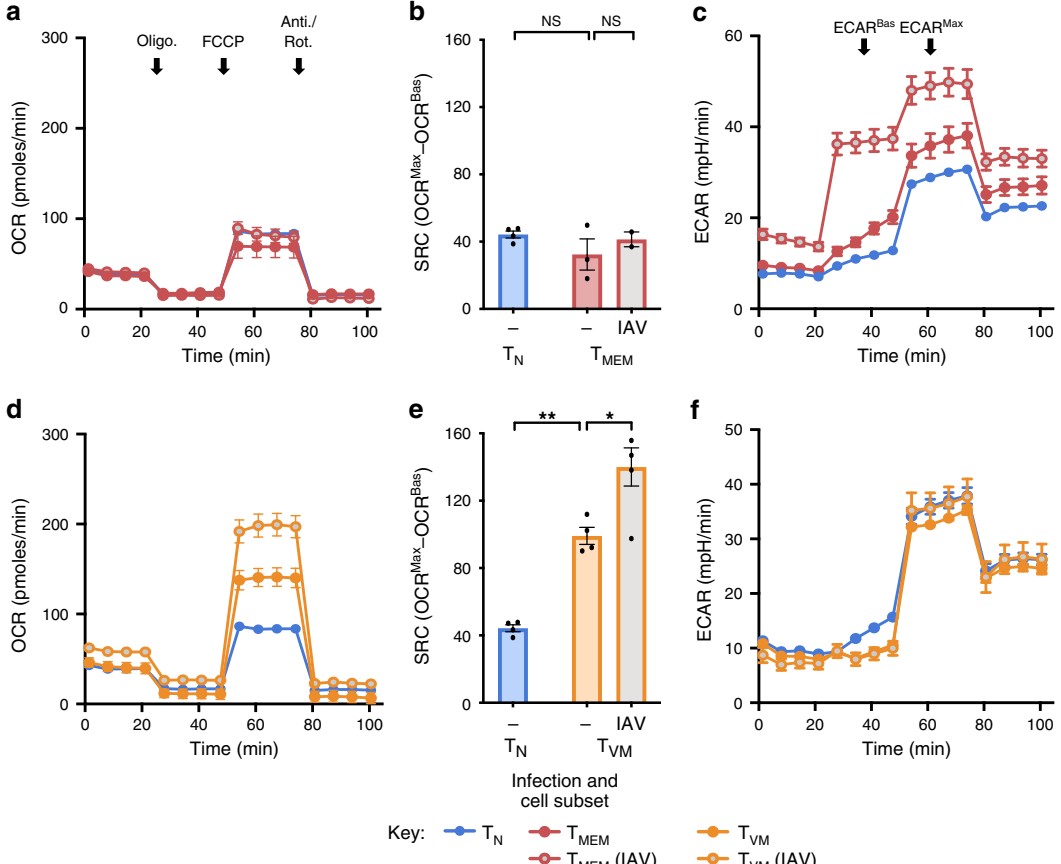

**Fig. 2 $T_{VM}$ cells increase SRC with recent infection. a** OCR for sorted $T_N$ and $T_{MEM}$ cells from the spleens of uninfected young SPF mice or Tetramer$^+$ $T_{MEM}$ (IAV) cells from IAV-infected mice (20 days post infection) and **b** change in OCR for each sorted subset ($n = 4$–2, 3 experimental replicates). **c** ECAR for sorted $T_N$ and $T_{MEM}$ cells from young uninfected mice or Tetramer$^+$ $T_{MEM}$ (IAV) cells from IAV-infected mice (20 days post infection), with $ECAR_{Bas}$ and $ECAR_{Max}$ indicated ($n = 2$–4, 3 experimental replicates). **d** OCR for sorted $T_N$ and $T_{VM}$ cells from young uninfected mice or $T_{VM}$ (IAV) cells from IAV-infected mice (20 days post infection) and **e** change in OCR for each sorted subset ($n = 4$–5, 3 experimental replicates). **f** ECAR for sorted $T_N$ and $T_{VM}$ cells from young uninfected mice or IAV-infected mice (20 days post infection) ($n = 5$, 3 experimental replicates). Shown is mean ± SEM. NS indicates not significant, * indicates $p \leq 0.05$, ** indicates $p \leq 0.01$, unpaired $t$ test.

The reciprocal analysis was also performed to determine the distribution of sorted $T_{VM}$ (CD44$^{hi}$CD49d$^{lo}$) and $T_{MEM}$ (CD44$^{hi}$CD49d$^{hi}$) cells from naive young, naive aged or LCMV-infected mice across classical $T_{CM}$ (CD44$^{hi}$CD62L$^{hi}$) and effector memory ($T_{EM}$; CD44$^{hi}$CD62L$^{lo}$) gates. When $T_{VM}$ cells were overlaid onto CD44/CD62L plots, they distributed predominantly to the $T_{CM}$ gate (~80%; Fig. 3b). $T_{MEM}$ cells were predominantly distributed to the $T_{EM}$ gate (49-95%; Fig. 3c), although a substantial proportion was found in the $T_{CM}$ gate in naive young mice. We also found that $T_{MEM}$ (IAV) cells were predominantly $T_{EM}$ cells (88.93 ± 6.67%) with some $T_{CM}$ cells (11.07 ± 6.67%) (Fig. 3d).

To definitively determine the relative SRC of $T_{CM}$ and $T_{EM}$ cells alongside $T_{VM}$ cells, we performed the Seahorse Mito Stress assay on $T_N$ and $T_{VM}$ cells isolated from naive SPF mice and $T_{EM}$ and $T_{CM}$ cells isolated from LCMV-infected mice at 60 days post infection. As previously observed, $T_{VM}$ cells had a significantly higher SRC than all other subsets (Fig. 3e, f). In addition, we found that $T_{EM}$ cells had a significantly lower SRC than all other subsets, and $T_{CM}$ cells had a modestly higher SRC than $T_N$ cells (Fig. 3e, f). This comprehensively demonstrates that high SRC is not a defining feature of conventional $T_{MEM}$ cells, in particular the $T_{EM}$ cell subset, but is instead characteristic of $T_{VM}$ cells.

**SRC correlates with markers of survival, not functionality.** High SRC had been proposed as a $T_{MEM}$ cell characteristic facilitating both their enhanced functionality and long-term survival[4,5]. Accordingly, we aimed to determine if high SRC in $T_{VM}$ cells and the SRC increase with age correlated with enhanced CD8$^+$ T cell functionality or survival.

Three key functions of CD8$^+$ T cells during an immune response are proliferation, cytokine production (particularly IFN-γ) and cytotoxicity. We recently found that $T_{VM}$ cells from aged individuals, with the highest SRC, have very low TCR-driven proliferative capacity, although the few that can respond still produce IFN-γ[17]. To further evaluate the functionality of CD8$^+$ T cell subsets, their cytotoxic capacity immediately ex vivo was assessed. Sorted $T_N$, $T_{VM}$ and $T_{MEM}$ cells from young and aged OT-I mice were used in an in vitro cytotoxicity assay with ovalbumin-loaded splenocytes as targets. In young mice, $T_{VM}$ cells were substantially more cytotoxic than $T_N$ cells and equivalent to $T_{MEM}$ cells (Fig. 4a). With increasing age, the cytotoxicity of $T_N$ cells remained low while the cytotoxicity of $T_{VM}$ and $T_{MEM}$ cells declined substantially (Fig. 4a). These data, along with our previous work[17], demonstrate that $T_{VM}$ and $T_{MEM}$ cells decline in cytotoxic and proliferative capacity with age, despite their increasing SRC.

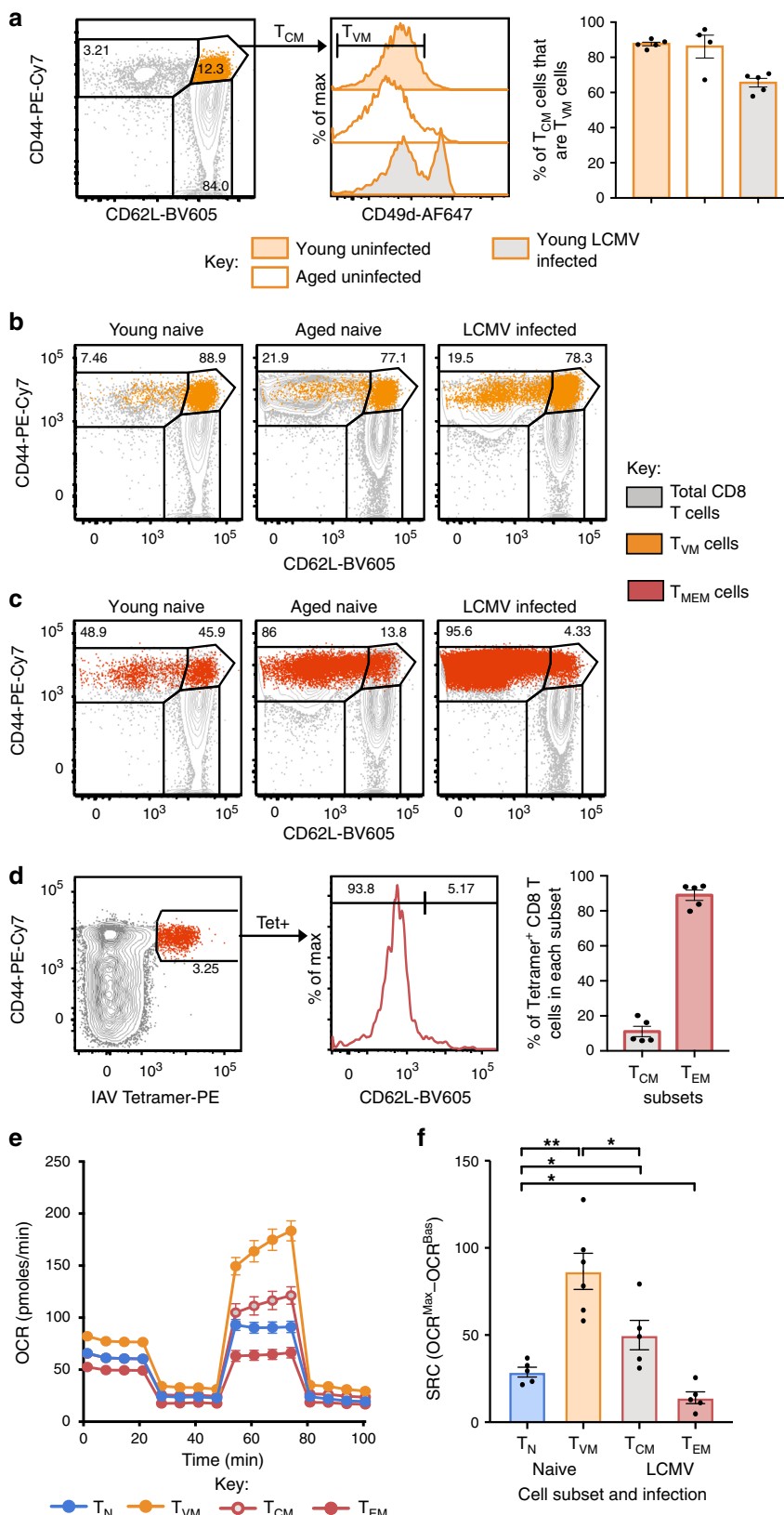

To formally test whether the capacity for any CD8[+] T cell function (proliferation, cytokine production or cytotoxicity) correlated with SRC, linear regression analyses were performed across the cell types and ages. There was no significant correlation observed for any of the functions (Fig. 4b–d), comprehensively demonstrating that high SRC does not necessarily indicate enhanced CD8[+] T cell function, particularly in CD8[+] T cells from aged individuals.

Our previous work indicated that $T_{VM}$ cells from aged individuals, which exhibit the highest SRC, had a survival advantage following adoptive transfer[17], which appeared to parallel expression of the anti-apoptotic protein, Bcl-2. Linear

**Fig. 3 $T_{VM}$ cells comprise the majority of the CD44$^{hi}$CD62L$^{hi}$ $T_{CM}$ cell population. a** Histograms for CD49d expression on CD44$^{hi}$CD62L$^{hi}$ CD8$^+$ T cells ($T_{CM}$ cells) from naive young mice, naive aged mice and young mice after infection with LCMV (40 days post infection), with bar graphs showing the proportion of $T_{CM}$ cells that are $T_{VM}$ cells ($n = 4$–5). **b** Overlays of CD44$^{hi}$CD49d$^{lo}$ $T_{VM}$ cells on total CD8$^+$ T cells with $T_{EM}$/$T_{CM}$ cell gating (CD44$^{hi}$CD62L$^{lo}$ and CD44$^{hi}$CD62L$^{hi}$, respectively). **c** Overlays of CD44$^{hi}$CD49d$^{hi}$ $T_{MEM}$ cells on total CD8$^+$ T cells. **d** Representative dot plots identifying IAV-specific tetramer$^+$ CD8$^+$ T cells that are CD62L$^{hi}$ ($T_{CM}$ cells) or CD62L$^{lo}$ ($T_{EM}$ cells) (60 days post infection), with bar graphs of the average frequency of tetramer$^+$ CD8$^+$ T cells that are in each subset ($n = 5$). **e** OCR for sorted $T_N$ and $T_{VM}$ cells from young uninfected mice and sorted $T_{CM}$ and $T_{EM}$ cells from LCMV-infected mice (60 days post infection) and **f** change in OCR for each sorted subset ($n = 5$–6, 3 experimental replicates). Data from **a–d** are representative of at least 2 individual experiments. Shown is mean ± SEM. * indicates $p \leq 0.05$, ** indicates $p \leq 0.01$, unpaired $t$ test.

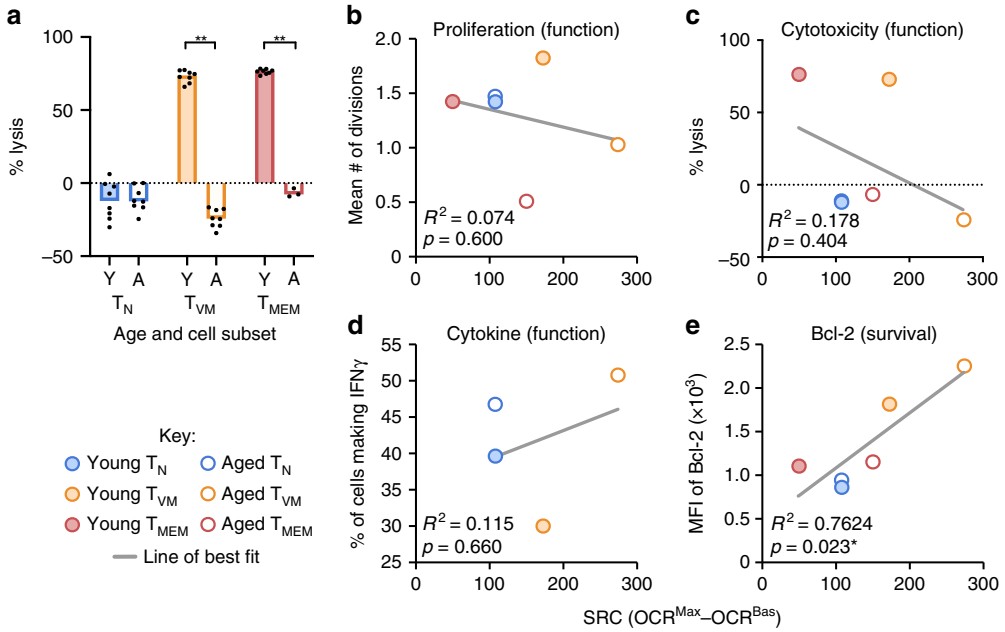

**Fig. 4 SRC correlates with Bcl-2 expression but not with CD8$^+$ T cell functionality. a** Percent lysis of OVA-loaded targets by sorted $T_N$, $T_{VM}$ and $T_{MEM}$ cells from young or aged OT-I mice ($n = 3$–8). * Indicates $p \leq 0.05$, ** indicates $p \leq 0.01$, unpaired $t$ test, data are representative of at least 3 individual experiments. Simple linear regression analyses of mean SRC from Fig. 1b against **b** the mean number of divisions of sorted CD8$^+$ T cells following 60 h TCR stimulation from ref. [17], **c** the average percent lysis from (**a**), **d** the average proportion of sorted CD8$^+$ T cells making IFN-γ at 36 h after TCR stimulation from ref. [17], and **e** the median fluorescence intensity (MFI) of Bcl-2 expression from ref. [17].

regression analysis of Bcl-2 expression vs SRC across the cell types and ages revealed a significant positive correlation (Fig. 4e), highlighting that SRC is associated with Bcl-2 expression as a surrogate for survival capacity in CD8$^+$ T cells.

**Increased $T_{VM}$ cell IL-15 signalling drives increased SRC.** Given the strong correlation of Bcl-2 with SRC, mediators of Bcl-2 expression were assessed to define the drivers of high SRC in $T_{VM}$ cells with age. The cytokine, IL-15 (reviewed in ref. [31]), was a strong candidate as it has been shown to promote survival of memory phenotype CD8$^+$ T cells through induction of STAT5 phosphorylation (pSTAT5) and expression of Bcl-2 (ref. [32]). To understand how IL-15 signalling might change with age in the different subsets, we assessed the expression of cytokine receptor subunits and downstream signalling in $T_N$, $T_{VM}$ and $T_{MEM}$ cells from young and aged mice.

Expression of IL-15Rβ was low on $T_N$ cells, modestly higher on $T_{MEM}$ cells, and markedly and significantly higher on $T_{VM}$ cells from young mice (5-fold, $p < 0.0001$; Fig. 5a). In addition, while neither $T_N$ nor $T_{MEM}$ cells exhibited substantial age-related changes in expression, $T_{VM}$ cells exhibited a marked increase in IL-15Rβ expression with age (Fig. 5a). When each subset was sorted and stimulated with soluble IL-15, the intensity of pSTAT5 tracked with receptor expression levels, with $T_{VM}$ cells from

young mice exhibiting high pSTAT5 that increased further with age (Fig. 5b). Together, these data indicate that, of all CD8$^+$ T cell subsets, $T_{VM}$ cells exhibit the greatest sensitivity to IL-15 signalling and this sensitivity increases with age.

IL-15 is regarded as a critical cytokine for both $T_{VM}$ cells and $T_{MEM}$ cells as evidenced by the fact that IL-15 knockout (KO) mice lose approximately half of their memory phenotype (CD44$^+$) cells[33] and the $T_{VM}$ cell population fails to develop in young mice in the absence of IL-15 or IL-15Rα[14,15]. To ascertain whether the IL-15Rβ expression and IL-15 responsiveness of CD8$^+$ T cell subsets reflects their relative dependence on IL-15, we assessed the development and persistence of each subset in young and aged IL-15 KO mice. $T_{VM}$ cells were absent in young IL-15 KO mice, consistent with previous studies, and $T_{VM}$ cells were also absent in aged mice (Fig. 5c, d). Of note, the $T_{MEM}$ cell population appears to be relatively intact in aged mice (Fig. 5c, d). This highlights that IL-15 is absolutely essential for the development of $T_{VM}$ cells, but is dispensable for the generation and maintenance of many $T_{MEM}$ cells, in aged mice.

IL-15 is produced at low levels in steady state but can be dramatically upregulated in DCs during infection in response to type I IFN signalling[34]. To test whether production of IL-15 during infection was leading to the observed increase in SRC in $T_{VM}$ cells (Fig. 2d, e), we administered an IL-15 neutralising

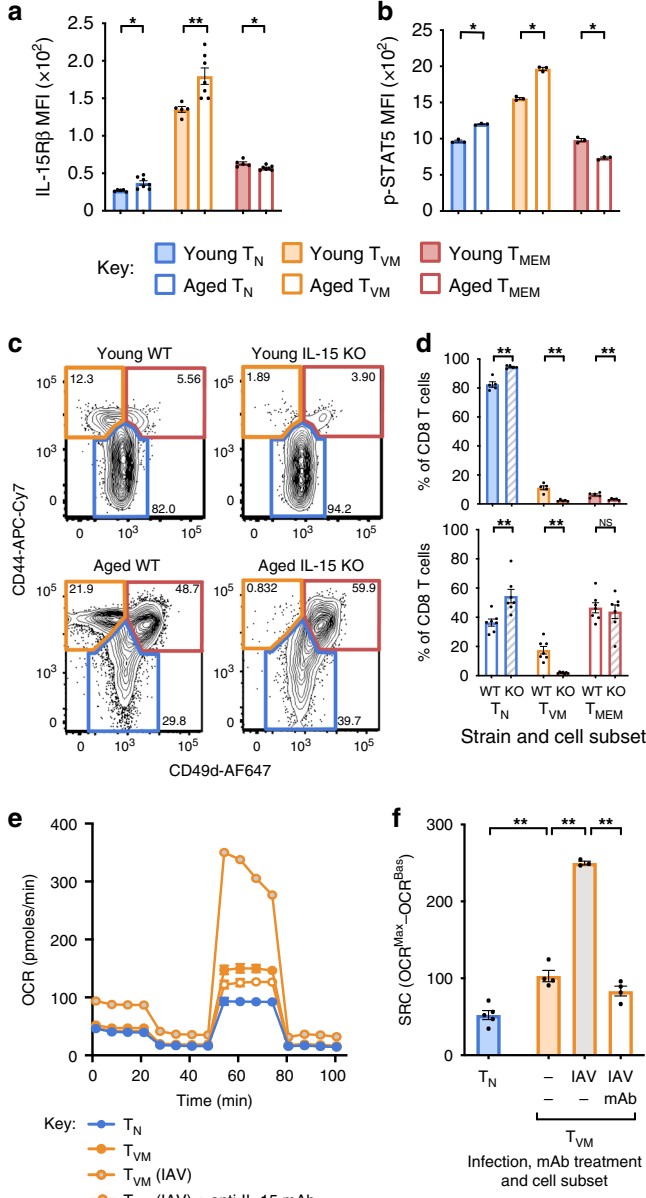

**Fig. 5 High IL-15 sensitivity in $T_{VM}$ cells increases with age and mediates increased SRC. a** IL-15Rβ MFI directly ex vivo from individual mice ($n = 5–7$) or **b** pSTAT5 MFI after 15 min of stimulation with IL-15 in vitro on sorted $T_N$, $T_{VM}$ and $T_{MEM}$ cells from young or aged mice ($n = 3$). **c** Representative dot plots for $CD8^+$ T cells gated on $T_N$ ($CD44^{lo}$), $T_{VM}$ ($CD44^{hi}CD49d^{lo}$) and $T_{MEM}$ ($CD44^{hi}CD49d^{hi}$) cells (frequency of $CD8^+$ T cells indicated) and **d** frequency of $CD8^+$ T cells in each subset in WT or IL-15 KO mice ($n = 5$ for WT and 7 for KO). **e** OCR for sorted $T_N$, $T_{VM}$ and $T_{MEM}$ cells from young uninfected, IAV-infected and IAV-infected/IL-15 neutralising mAb treated mice and **f** change in OCR for each sorted subset. Shown is mean ± SEM. \* Indicates $p \leq 0.05$, \*\* indicates $p \leq 0.01$, unpaired $t$ test, data for **a**, **b**, **e**, **f** are representative of at least 3 individual experiments.

monoclonal antibody (mAb) to young mice infected with IAV. The mAb was administered at days 0 and 3 after infection and SRC was assessed at day 14. Abrogating IL-15 signalling during infection was sufficient to abrogate the infection-driven increase in SRC in $T_{VM}$ cells (Fig. 5e, f). Collectively, these data highlight that the metabolic profile, and in particular elevated SRC, that was previously associated with $T_{MEM}$ cell function and longevity,

appears to be a direct function of IL-15 exposure and independent of $T_{MEM}$ phenotype. In addition, the sensitivity of $T_{VM}$ cells to IL-15 appears to account for the high SRC observed in these cells directly ex vivo from young mice and the increase in SRC in $T_{VM}$ cells with age.

**Elevated IL-15Rβ and SRC in older human $CD8^+$ T cells.** To determine whether our findings in mice were relevant to humans, we first analysed expression of IL-15Rβ on young adult (20–30 yo) and older (60–80 yo) human $CD8^+$ T cells. We found significantly higher expression of IL-15Rβ on $T_{VM}$ cells compared to $T_N$ cells in young adults, and this expression increased significantly with age across all subsets (Fig. 6a, b), similar to our observations in mice. Moreover, we observed a trend toward increased SRC in total $CD8^+$ T cells with advanced age ($p = 0.1$) (Fig. 6c, d). Given our previous description of diminished proliferative capacity in $CD8^+$ T cells from older humans[35], these data suggest a similar lack of correlation between SRC and functionality in human $CD8^+$ T cells. Moreover, these data indicate a correlation between elevated SRC and an age-related increase in IL-15 sensitivity of human $CD8^+$ T cells.

## Discussion
By detailed dissection of the metabolic phenotype of $T_{VM}$ cells compared with $T_N$ and $T_{MEM}$ cells, we have demonstrated that high SRC correlates best with IL-15 sensitivity and Bcl-2 expression, and it is therefore selectively associated with the superior survival capacity of $T_{VM}$ cells. In contrast, neither elevated SRC nor substantial IL-15 dependence were characteristics of conventional antigen-driven memory $CD8^+$ T cells, and SRC did not associate with multiple measures of T cell functionality across various cell types and ages. Our data, therefore, illustrate that previous work defining high SRC as a characteristic of $T_{MEM}$ cells, and $T_{CM}$ cells in particular, is likely due to the conflation of these populations with $T_{VM}$ cells, as well as the use of high levels of IL-15 to generate the memory-phenotype $CD8^+$ T cells studied. As previously mentioned, $T_{VM}$ cells cannot be distinguished from $T_{CM}$ cells without inclusion of the marker for $CD49d^{20}$. The original study proposing SRC as a characteristic of $T_{MEM}$ cells either isolated $CD8^+CD44^{hi}CD62L^{hi}$ cells ex vivo from *Listeria monocytogenes* infected mice or generated cells in vitro using high levels of exogenous IL-15 (ref. [4]). Subsequent work proposed that high SRC partitioned preferentially with the $T_{CM}$ cell population but $T_{CM}$ cells in this study also appeared to be identified as $CD8^+CD44^{hi}CD62L^{hi}$ cells from LCMV-infected mice[7]. Based on our findings, we contend that these in vivo strategies would have included a substantial population (60–80% of sorted cells) of $T_{VM}$ cells (Fig. 3a), while the in vitro strategy, which has been widely used to define phenotypic, functional and metabolic characteristics of memory cells[3–6,8,36,37], relies on robust IL-15 signalling to drive a memory-like phenotype. Recent work has suggested that IL-15 signalling has direct effects on metabolic profiles, with vaccine-induced effector $CD8^+$ T cells shown to depend predominantly on IL-15 for increased SRC[38], and another study showing increased expression of Bcl-2, increased IL-15Rβ and metabolic adaptations in memory $CD8^+$ T cells ($CD44^{hi}$) from aged mice, again incorporating $T_{VM}$ cells[23]. IL-15 signalling is also known to promote OXPHOS in other cell types, with IL-15 overexpression driving oxidative metabolism in both adipose tissue and skeletal muscle[39,40]. Thus, we contend that the selective sensitivity of $T_{VM}$ cells to IL-15 is the basis for the uniquely elevated SRC in those cells. Our data suggest that physiological IL-15 signalling can drive high SRC in $T_{VM}$ cells, and that this can be further increased with recent infection and increasing age. Accordingly, we propose that high SRC is an indicator of recent

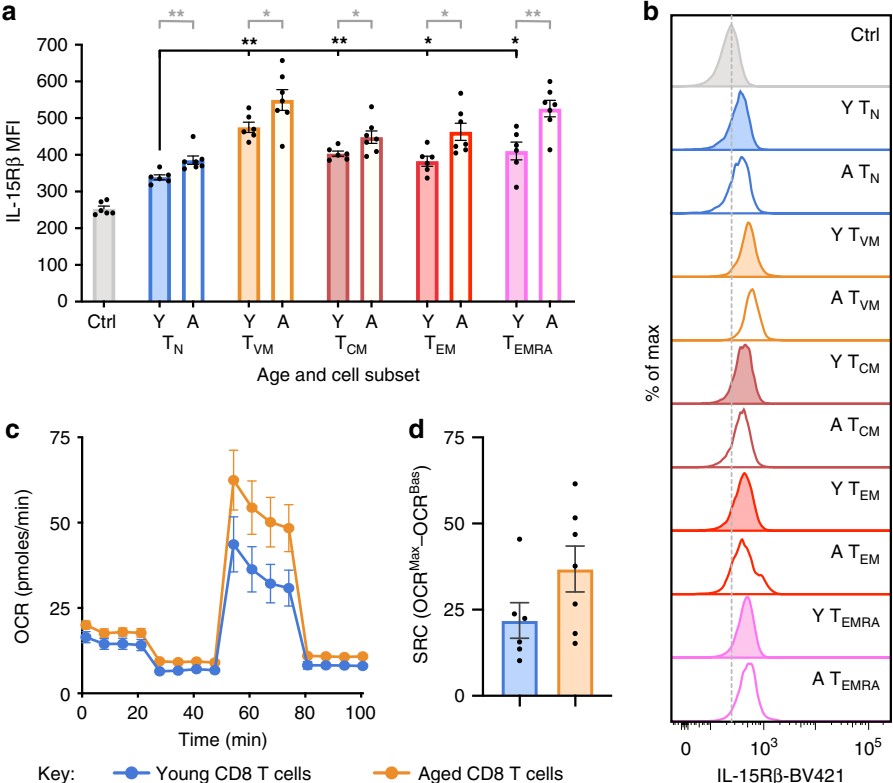

**Fig. 6 Increased IL-15Rβ expression and SRC in older human CD8+ T cells. a** IL-15Rβ MFI directly ex vivo on CD8 T cell subsets from individual young (20–30 yo) or older (60–80 yo) adult human donors ($n = 6$–7) and **b** Representative histograms of IL-15Rβ expression, with expression on CD4+ T cells used as a control. **c** OCR for enriched CD8+ T cells from young or older human donors and **d** change in OCR for young or older human donors. This experiment was performed once. Bars or datapoints represent mean ± SEM. * Indicates $p \leq 0.05$, ** indicates $p \leq 0.01$, unpaired $t$ test.

IL-15 signalling in T cells, which itself appears to be a defining characteristic of naturally generated $T_{VM}$ cells, rather than $T_{MEM}$ or $T_{CM}$ cells.

This observation also raises the question of whether other well-accepted characteristics of $T_{MEM}$ populations are primarily canonical $T_{VM}$ cell characteristics[20], particularly with regard to IL-15. The dependence on IL-15 for survival is a well-accepted characteristic of antigen-experienced $T_{MEM}$ cells, alongside increased levels of IL15Rβ, and sensitivity to IL-15 stimulation[33,41]. Here, we demonstrate that $T_{VM}$ cells express higher levels of receptor, are more sensitive to IL-15 signalling, and are entirely dependent on IL-15 for their generation and maintenance, while $T_{MEM}$ cells are surprisingly resistant to a lack of IL-15. Furthermore, IL-15 has been traditionally regarded as a key mediator of CD4 help for antigen-specific CD8+ T cell responses, with CD4 T cells providing help by engaging through CD40L/CD40 interactions with DCs to amplify their IFN-induced production of IL-15 (ref. [42]). The provision of CD4 help, including type I IFN and IL-15, during an antigen-specific CD8+ T cell response limits CD8+ T cell contraction, resulting in a memory population responsive to secondary antigenic stimulation[41]. However, these cytokines also directly expand memory-phenotype populations in an antigen non-specific manner[34,43] and are crucial for generation of $T_{VM}$ cell populations[14,15]. To better delineate the qualitative and quantitative impact of IL-15 on antigen-experienced $T_{MEM}$ vs antigen-naive $T_{VM}$ cells, it is imperative to perform more nuanced analyses using *bona fide* antigen-experienced $T_{MEM}$ cell populations. This distinction is more than semantic; it is essential to accurately define the fundamental effects of IL-15 on T cell populations and to determine whether treatments using IL-15 or downstream targets will primarily act upon antigen-induced memory T cell populations or for the improved recruitment of antigenically naive CD8+ T cells following primary antigen encounter.

Of note, this data highlights that high SRC is not necessarily linked to superior CD8+ T cell function. Previously, high SRC was proposed to facilitate accelerated proliferation and increased cytokine production with in vitro generated $T_{MEM}$ cells[5]. We and others have previously demonstrated that proliferative capacity is highest in $T_{VM}$ cells, followed by $T_N$ and $T_{MEM}$ cells directly ex vivo in young individuals[9,17,19,22], which tracks with their SRC. However, any correlation breaks down with ageing, as proliferative capacity is highest in $T_N$ cells and relatively poor in $T_{VM}$ and $T_{MEM}$ cells[17,22] and the latter two populations exhibit elevated SRC. We propose that this lack of correlation between SRC and function in aging is compounded by an age-related defect in TCR signalling that uncouples metabolic potential in resting cells from metabolic engagement and functionality after activation. Dysfunctional TCR-driven proliferation in $T_{VM}$ cells from aged individuals and dysregulation of signalling cascades downstream of TCR engagement are evident with age[17,22,44,45]. This dysregulation may prevent $T_{VM}$ cells from efficiently engaging any increased mitochondrial reserve upon TCR signalling. To test this paradigm further, it would be informative to assess SRC in other T cell subsets, such as stem cell memory T cells, which are highly proliferative, and resident memory T cells, which retain a level of proliferative capacity in situ and are highly IL-15 dependent. Unfortunately, the low frequency of these cells precludes their analysis using the current Seahorse technology.

Interestingly, our mouse data appears to partly contradict earlier work suggesting that human naive CD8+ T cells exhibit an age-related reduction in SRC, although an age-related increased

mitochondrial load was observed[46]. This study used older individuals up to the age of 85 yo, which may be older than the equivalent mouse age of 18–20 mo characterised here. Thus, it is possible that the SRC is reduced in $T_N$ cells at extreme ages. Irrespective, our results in mice highlight that a dramatic metabolic shift occurs within the CD8$^+$ T cell population from aged mice, coincident with our analyses of dysfunction and leading to a substantial increase in SRC in memory-phenotype cells. This also appears to be supported by our observation of elevated SRC in older human CD8$^+$ T cells.

Our data, and others, suggests that high SRC might be a more relevant indicator of T cell longevity, rather than functionality. The elevated SRC observed in $T_{VM}$ cells is paralleled by our previous data suggesting these cells have a survival advantage both in vivo and in vitro[13,17]. Moreover, IL-15 is necessary for both elevated SRC and essential for $T_{VM}$ cell survival[15]. Finally, SRC has been found to be directly responsible for survival of myocytes under conditions of hypoxia or nutrient deprivation[47]. We propose that the differential sensitivity of distinct CD8$^+$ T cell subsets to IL-15 exacerbates its effects in states of increased IL-15 production, such as infection or ageing. IL-15 is produced at low levels in the steady state but it can be dramatically upregulated in DCs during infection, in response to type I IFN signalling[34]. Type I IFN and IL-15 signalling may also increase with age, as a result of heightened inflammation, or inflammageing[48]. Increased IL-15 transcription has been observed in bone marrow[49] and lymph node stromal cells with age[50]. Bcl-2 was similarly seen to be increased in CD8$^+$ T cells that survived after transfer into a lymphopenic environment, which also correlated with metabolic shifts[51]. While SRC and IL-15 sensitivity are clearly linked, the mechanism by which IL-15 signalling drives increased SRC remains unclear. It may upregulate Bcl-2 - Bcl-2 family members have been shown to increase mitochondrial capacity by increasing CIV expression[52] and inhibiting mitophagy[53]. It remains unclear, however, whether there is a causal link between SRC and survival or whether these are independently mediated by IL-15. Additionally, while our results show that IL-15 is necessary for high SRC in $T_{VM}$ cells, it may not be entirely sufficient, and additional signalling pathways may be required to co-ordinate with IL-15 to drive increased SRC.

In the T cell field, mitochondrial fusion has been largely associated with advantageous outcomes; namely elevated SRC and memory T cell formation[3,6], although this association was not observed in this study, while a more fragmented (or fissed) mitochondria is associated with effector T cells[6]. However, while fission of mitochondria is essential for cytokinesis and for isolating dysfunctional mitochondria to allow autophagic degradation[54], fusion of mitochondria is generally considered to reflect an age- or senescence-related stress response to promote survival of the cell by enabling functional complementation, whereby damaged or dysfunctional mitochondria fuse with intact mitochondria, to maintain ATP production and functionality[55,56]. We speculate that mitochondrial fusion in this biological scenario reflects an age-related functional complementation.

Collectively, our study underlines how CD8$^+$ T cell subset function and survival is controlled by molecular and metabolic pathways, knowledge of which is essential for design of interventions that restore functionality or modify survival in T cells from aged individuals.

## Methods
**Mice.** Female C57BL/6 mice and male CD45.1+ OT-I mice on a C57BL/6 background were bred and housed in specific-pathogen-free (SPF) conditions at the Monash Animal Research Platform (MARP) and Animal Research Facility (ARL) at Monash University. IL-15 knockout (KO) mice were housed in SPF conditions at the Biomedical Research Facility in the Department of Microbiology and

Immunology (DMI) at the University of Melbourne. Young mice were defined as 2–3 months old (mo), and aged mice were ex-breeding stock that were 18–20 mo, unless otherwise noted in the figure legends. During tissue harvests, mice were examined for gross abnormalities, tumours, and enlarged lymph nodes and spleens, and excluded from analyses if these were evident. All animal experimentation was conducted following the Australian National Health and Medical Research Council Code of Practice for the Care and Use of Animals for Scientific Purposes guidelines for housing and care of laboratory animals and performed in accordance with Institutional regulations after pertinent review and approval by the University of Melbourne and Monash University Animal Ethics Committees.

**Seahorse Mito Stress assay.** A Seahorse XFe96 Bioanalyser (Agilent) was used to determine OCR and ECAR for sorted CD8$^+$ T cell subsets. Sorted cells were washed in assay media (XF Base media (Agilent) with glucose (10 mM), sodium pyruvate (1 mM) and L-glutamine (2 mM) (Gibco), pH 7.4 at 37 °C) before being plated onto Seahorse cell culture plates coated with Cell-Tak (Corning) at 2×10$^5$ cells per well. After adherence and equilibration, cell OCR and ECAR was measured during a Seahorse Mito Stress assay (Agilent), with addition of oligomycin (1 μM), carbonyl cyanide 4-(trifluoromethoxy) phenylhydrazone (FCCP; 1.2 μM) and antimycin A and rotenone (0.5 μM each)). To assess pathways contributing to SRC, we added etomoxir at either 200 or 5 μM (Fig. 1h) or 500 mM of 2-deoxy-D-glucose (Supplementary Fig. 2), after FCCP and prior to antimycin A/rotenone. Assay parameters were as follows: 3 min mix, no wait, 3 min measurement, repeated 3–4 times at basal and after each addition. SRC was calculated as oxygen consumption rate (OCR) at maximum rate (OCR$_{Max}$) − OCR in basal state (OCR$_{Bas}$).

**Electron microscopy.** To prepare for transmission electron microscopy, at least $1 \times 10^6$ sorted CD8$^+$ $T_N$, $T_{VM}$, and $T_{MEM}$ cells from young and aged mice were fixed in 2.5% glutaraldehyde in 100 mM sodium cacodylate buffer for 2 h, washed in cacodylate buffer, and post-fixated (1% osmium tetroxide, 1.5% potassium ferricyanide, 65 mM sodium cacodylate buffer) for 1 h before storage in cacodylate buffer. Samples were serially dehydrated with increasing concentrations of ethanol, then propylene oxide before being embedded in serial ratios (3:1, 2:1, 1:1, 1:2 and 1:3) of propylene oxide:epon araldite resin, with samples in 100% epon araldite resin polymerised at 60˚C for 48 h. Embedded blocks were microtomed into 60 nm sections that were stained with 2.5% uranyl acetate for 15 min then Reynold's lead citrate for 3 min. Sections were imaged on a JEOL JEM-1400 electron microscope operated at 80 kV using a sCMOS Matataki Flash camera.

**Confocal microscopy.** To prepare for confocal microscopy, at least $1 \times 10^6$ sorted $T_N$, $T_{VM}$, and $T_{MEM}$ cells from young and aged mice were seeded on 0.1% gelatin-coated Fluorodishes (World Precision Instruments), allowed to settle, adhered for 30 min and fixed with 4% (w/v) paraformaldehyde in PBS (pH 7.4) for 10 min. After permeabilisation with 0.5% (w/v) Triton X-100 in PBS, cells were incubated with primary antibody against Cytochrome C (CytC, mouse monoclonal, BD Biosciences, 556432; 1:500 in 3% BSA-1xPBS) for 60 min at room temperature. The primary antibody was labelled with Alexa-Fluor-488 conjugated anti-mouse-IgG (Molecular Probes, 1:500 in 3% BSA-1xPBS). Hoechst 33258 (1 μg/ml) was used to stain nuclei.

Confocal microscopy was performed on a Leica TCS SP8 confocal microscope (405 nm, 488 nm, 552 nm, 647 nm; Leica Microsystems) equipped with HyD detectors using a 63×/1.40 NA oil immersion objective (HC PLAPO, CS2, Leica Microsystems). Microscopy data was recorded using the Leica LAS X Life software. Images in all experimental groups were obtained using the same settings. Z-sectioning was performed using 150-nm slices. Leica.lif files were converted to multi-colour.tiff composite stacks using custom-written Fiji/ImageJ macros (Version 1.52n).

Images were analysed using Fiji and custom-written macros. Mitochondrial network area (footprint) was evaluated using the MINA plugin for Fiji based on maximum intensity projections of 3D-image stacks. Mitochondrial organelle morphology quantitation into three subcategories (fragmented, intermediate, fused) was evaluated manually based on confocal z-stacks from three independent experiments totalling to 140 cells. All graphical representations and statistical analysis were carried out on Prism (v7.0a, GraphPad) using two-way ANOVA or Student's t-tests. For representational figures, images were median filtered (1px) using ImageJ and Fiji.

**Blue-native PAGE.** To assess expression levels of ETC components, $5 \times 10^5$ sorted $T_N$, $T_{VM}$ and $T_{MEM}$ cells from young and aged mice were solubilized in 1% digitonin solubilization buffer, subjected to BN-PAGE and transferred to PVDF membrane[57] before immunoblotting for COX5A. Complex IV was detected using antibodies to COX5A (Santa Cruz sc-376907) and horseradish peroxidase coupled secondary antibodies and ECL chemiluminescent substrate (BioRad) were used for detection on a BioRad ChemiDoc XRS+ imaging system. The PVDF membrane was also stained with Coomassie Blue (50% methanol, 7% Acetic acid, 0.05% Coomassie Blue R) to assess relative protein loading.

### Table 1 Human PBMC donor details.

| Young Adults (20-30 years old (yo)) | | Older Adults (60-80 yo) | |
|---|---|---|---|
| **Sex** | **Age** | **Sex** | **Age** |
| Female | 23 | Male | 69 |
| Female | 22 | Female | 74 |
| Female | 20 | Female | 70 |
| Male | 27 | Female | 74 |
| Male | 22 | NR | 79 |
| Female | 24 | NR | 79 |
| | | NR | 62 |

*NR* not recorded.

**Identification, sorting and phenotyping of T cell subsets.** For identification and isolation of mouse $T_N$, $T_{VM}$, and $T_{MEM}$ cells, samples were processed and subsets were identified using the staining panel and gating strategy described previously[17]. For sorting, $T_N$ cells were defined as CD44lo (the bottom 30% of CD44 expression based on gating in a young, untreated control mouse); CD44int cells were not included in sorted populations.

For characterisation of surface expression of CD69, CD5, CD127 and CD122 on mouse subsets, the following panel was used: LIVE/DEAD Fixable AquaBlue Viability Dye (Life Technologies), anti-Dump (B220, CD4, CD11c, CD11b, F4/80, NK1.1)-FITC (BD Pharmingen; all 1:400), anti-CD8-PacBlue (53-6.7; BD Pharmingen; 1:200), anti-CD49d-AF647 (R1-2; Biolegend; 1:400), anti-CD44-APC-Cy7 (IM7; Biolegend; 1:400) and either anti-CD69:PE (H1:2F3; Biolegend; 1:400), anti-CD5:PE (53-7.3; BD Pharmingen; 1:200), anti-CD127:PE (A7R34; eBioscience; 1:400), or anti-CD122:PE (TM-β1; BD Pharmingen; 1:400).

For overlays of $T_{CM}/T_{EM}$ gating with $T_N$, $T_{VM}$ and $T_{MEM}$ subsets, the following panel was used: LIVE/DEAD Fixable Near IR Viability Dye (Life Technologies), anti-Dump (B220, CD4, CD11c, CD11b, F4/80, NK1.1; all 1:400)-FITC (BD Pharmingen), anti-CD8-BUV395 (53-6.7; BD Pharmingen; 1:400), anti-CD49d-AF647 (R1-2; Biolegend; 1:400), anti-CD44-PE-Cy7 (IM7; Biolegend; 1:1000) and anti-CD62L:BV605 (MEL-14; BD Pharmingen; 1:400). PE-labelled IAV-specific tetramers were included in Fig. 2 and Fig. 3d and were a pool of H2-D^b-based tetramers loaded with $NP_{366}$, $PA_{224}$ and $PB1-F2_{62}$ epitopes.

**IAV infection, LCMV infection and IL-15 neutralisation.** For influenza A virus (IAV) infection, mice were anesthetized by isoflurane inhalation and infected intranasally with $1 \times 10^4$ plaque-forming units of the HKx31 (H3N2) IAV strain in 30 μL of PBS and spleens were harvested at indicated timepoints.

For lymphocytic choriomeningitis virus (LCMV) infection, mice were administered 3000 plaque-forming units (PFU) of LCMV (strain WE) intravenously and spleens were harvested at indicated timepoints.

For IL-15 neutralisation, 25 μg of an IL-15/Ra neutralising monoclonal antibody (mAb) (GRW15PLZ; eBioscience) was administered intraperitoneally on day 0 immediately prior to infection and at day 3 after IAV infection.

**Cytotoxicity assays.** Effector $T_N$, $T_{VM}$ and $T_{MEM}$ cells were sorted from young and aged male CD45.1+ OT-I mice. Target splenocytes from male C57BL/6 mice were stained with intermediate or high concentrations of Cell Trace Violet (CTV, Molecular Probes) and left unloaded (CTV low) or loaded with Ovalbumin-derived peptide (SIINFEKL) at 0.1 μM (CTV high). Unloaded and Ova-loaded targets were mixed in equal proportions with beads (1:1:1) and then effector cells were added at ratio of 5:1 with Ova-loaded target cells. Cultures were incubated overnight at 37 °C in 5% CO₂. To identify live cells and differentiate effector cells from target cells, the sample was stained with Propidium Iodide (Molecular Probes), anti-CD45.1:APC-Cy7 (A20; Biolegend; 1:400) and anti-CD45.2:PE (104; Biolegend; 1:400). The ratio of live loaded target cells to beads after incubation with each effector cell type was normalised back to the ratio of live loaded target cells to beads in samples that were not incubated with effectors, to calculate the % lysis for each effector cell type.

**Phosphorylation assays.** $T_N$, $T_{VM}$ and $T_{MEM}$ cells were sorted and left unstimulated, or stimulated with IL-7 (10 ng/mL) or IL-15 (100 ng/mL) in cRPMI at 37°C in 5% CO₂ for 15 min. Cells were processed with Lyse/Fix solution and Perm Buffer II (BD Biosciences) before staining with anti-phospho-STAT5 (CST; 1:400) followed by phycoerythrin (PE)-conjugated anti-rabbit secondary mAb (CST; 1:500).

**Human analyses.** Human experimental work was conducted according to the Declaration of Helsinki Principles and to the Australian National Health and Medical Research Council (NHMRC) Code of Practice. Signed informed consent was obtained from all blood donors before the study. The study was approved by

the University of Melbourne Human Ethics Committee (HREC 1443389). Donor details are shown in Table 1.

Briefly, PBMCs were defrosted and rested in complete RPMI overnight. One sample was taken and stained[35] to identify CD8+ T cell subsets ($T_N$, $T_{VM}$, $T_{CM}$, $T_{EM}$ and $T_{EMRA}$) based on CD45RA, CD27, Pan-KIR and NKG2A expression, along with anti-human IL-15Rβ-BV421 (TU27; Biolegend; 1:200) and Propidium Iodide was substituted as the viability dye. The remaining sample was negatively enriched for CD8+ T cells using the human CD8+ T Cell Isolation Kit (Miltenyi Biotec) as per manufacturer's instructions, plated at $2 \times 10^5$ cells per well and run in a standard Seahorse Mito Stress assay, as detailed above.

**Statistical analyses.** Data were analysed in Graphpad Prism (version 7.0a) using the unpaired, two-tailed $t$-test without correction for multiple comparisons, as indicated in figure legends.

**Reporting summary.** Further information on research design is available in the Nature Research Reporting Summary linked to this article.

## Data availability

RNASeq data accessed during this study are available at GEO under the accession code GSE112304. All other data that support the findings of this study are available from the corresponding authors upon request. Raw data for all figures are provided in the Source data file. Source data are provided with this paper.

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

## Acknowledgements

We thank Adam Costin and Joan Clark for technical assistance in electron microscopy, and staff at Monash University Ramaciotti Centre for Cryo-Electron Microscopy, Monash University FlowCore, Monash Micro Imaging, and Monash University Animal Facilities. This work was supported by a Sylvia and Charles Viertel Senior Medical Research Fellowship, an Australian Research Council (ARC) Future Fellowship FT170100174, a National Health and Medical Research Council (NHMRC) Program grant APP1071916 (to N.L.L.), a Rebecca L. Cooper Foundation Medical Research Grant (to K.M.Q), a Viertel-Belberry Senior Medical Research Fellowship (to K.L.G.-J.), Monash Graduate Scholarship and Monash International Postgraduate Research Scholarship (to T.H.) a Monash University Biomedicine Discovery Scholarship (to L.C.) and funding from the Bonn and Melbourne Research and Graduate School (GRK2168). C.E.S. has received funding from the European Union's Horizon 2020 research and innovation programme under the Marie Skłodowska-Curie grant agreement No. 792532 and University of Melbourne McKenzie Fellowship laboratory support. KK is supported by a NHMRC Senior Research Fellowship Level B (GNT#1102792).

## Author contributions

K.M.Q. and N.L.L. designed the study, K.M.Q. performed the majority of experiments and analysed results, T.H. performed assays of T cell function, F.K. performed confocal microscopy, L.M.A. performed cytotoxicity assays, L.E.F. performed western blots, W.K.L. and G.R. performed electron microscopy, L.C., E.W-J., L.L., C.E.S, and K.K. contributed to the procurement and processing of samples, E.C.S. and M.J.D. optimised Seahorse XFe96 experiments, N.L.L., K.M.Q., M.T.R., K.G-J, L.M., and M.J.M. supervised the research, K.M.Q. and N.L.L. wrote the initial draft of the paper and all authors participated in writing the final paper.

## Competing interests

The authors declare no competing interests.
