## [Peer Review File · Nature Communications]

Reviewers' comments:

Reviewer #1 (Remarks to the Author):

Quinn et al. are following previous work from their own lab and other groups in the determination of the metabolic signature of T cell subsets. They elegantly show the need for further analysis of T cell subsets to define the corresponding metabolic status and flexibility of subsets such as Naive, Virtual memory and MEM CD8+ T cells in mice. The work is sound and brings novel to another level that was incomplete from previous studies. This includes the link between SRC, cell longevity and the role of IL-15 and Bcl2 in this process.

While the authors are correct that inclusion of further markers (such as CD49d) is important it is also important they discuss their own limitation. For instance, the study of ageing T cell in mice in relation to naive T cell as a precursor to other memory cells should take into account mice are not susceptible to thymic involution and that in humans, a pool of recent thymic emigrant T cells (RTE) are different metabolically from the pool of naive T cell with no antigenic-experience but with homeostatic proliferation experience. For instance, the interpretation "This study used individuals up to the age of 85 yo, which is significantly older than the human CD8+ T cells we have previously characterised (60-70 yo)¹⁷ and likely older than the equivalent mouse age of 15-18 mo characterised here." is flawed.

Also, the study dismissed the population of T Stem Cell Memory, which was shown to have the higher proliferative capacity among all T cells.

Expression of IL-15R subunits should be measured in all subsets and in the context of aging, in human T cells to at least partially validate the mechanisms in human.

The study should measure the expression of co-stimulatory molecules such as Cd28 and Cd27 as well as other well-accepted markers of T cell differentiation. For instance, Cd28 was shown to lead to Akt activation which in turn is involved in glycolysis. This is important in the context of the metabolic hypothesis.

This study focusses on the phenomenon of mitochondrial fusion but dismissed the fission. Both should be included and discussed.

Reviewer #2 (Remarks to the Author):

This study by Quinn et al. aims to address the notion that increased SRC is intimately linked to increased CD8 T cell function, a suggestion made, like much of the widely accepted immunometabolism data, from in vitro generated cells and confounded population identifications. This is a very important and well performed body of work that, primarily, re-attributes increased SRC among CD8 T cells to the virtual memory (rather than traditional Ag stimulated memory). This work uses direct ex vivo analysis and not only delineates and breaks apart a widely accepted generalizations regarding memory T cells and SRC, mitochondrial morphology and fatty acid oxidation, but provides a starting point from which to reassess what has been proposed in immunometabolism. It has been long known that memory CD8 T cells generated in different infection systems can result in distinct outcomes; why the field has accepted generalizations regarding metabolism from in vitro generated effector and memory T cells remains unclear, and this paper aids us in calling those findings into question. This is a foundation from which we can truly approach the nuanced field of T cell memory with an interest in metabolism. That said, the authors correlate IL-15 dependence and increased SRC, while this is the case for their population of virtual memory cells, I would argue that the authors have demonstrated that IL-15 is necessary for SRC in virtual memory cells (an IL-15 dependent population), however they have not demonstrated that IL-15 is sufficient for SRC increase. In the interest of avoiding new

mis-generalizations in the field, I would encourage the authors to either, re-calibrate the language to be very clear about what is and is not shown but correlated or evaluate other populations of IL-15 dependent cells for SRC. The latter would prove highly difficult, though Thomas Kupper has demonstrated feasibility. For example, TRM cells from various organs after LCMV have been demonstrated to be IL-15 dependent or independent (Schenkel & Fraser et al. 2016), do these distinctions correlate with increased SRC?

--Finally, the breakdown of mitochondrial morphology holding some bearing on function and etomoxir dosing are very important points of this paper.

Major Concern: Mentioned above regarding IL-15 and SRC being correlated in a IL-15 dependent population as required for SRC, however not necessarily sufficient for SRC increase. Explore further or adapt language accordingly.

-Major focus on spare respiratory capacity, however it is unclear, is this a physiologically relevant function or just a correlate? Please clarify within the text.

-The antigen depot known to be associated with IAV makes the day 20 timepoint of 'memory' cells in figure 2B difficult to interpret. I would recommend either using LCMV infected cells and memory at a d30 time point or later date after IAV. The increase in glycolysis but not OCR indicates this time point may simply be too early and these are indeed confounded with an effector population.

Minor Concerns:

-Please state in the text or legend where TMEM cells are derived from in Figure 1 & 2. It is stated but not until line 180 at a transitional point. Reader should not have to search different sections for this information.

-Authors state that the SRC tracked closely with CIV. This does not appear to be the case in a generalizable way. (Fig 1g & 1a-b) Specifically, TN cells demonstrated an increase in CIV between young and aged mice but have no increase in SRC.

-Line 76, CD44 should not be described as a marker of memory, necessarily, because it comes up early during activation as well.

-Figure 1B would be clearer to call the y-axis, SRC (OCRmax-OCRbasal).

Reviewer #3 (Remarks to the Author):

This is an exceptionally written, experimentally concise and compelling manuscript detailing the metabolic profiles of naive and memory T cell subsets, in young and aged mice. In the process, it details some very important findings regarding the role of mitochondrial metabolism and FAO in the regulation of antigen-experience and -inexperienced T cell memory survival. The data they present continues to shake up the existing paradigm that is based almost exclusively on highly contrived in vitro models and inappropriate usage of metabolic inhibitors to promote the model that memory T cells have elevated SRC as a result of elevated FAO. The present manuscript punches holes in both of these assertions, and in a fashion that cannot be easily disregarded. Collectively, the conclusions of the manuscript are for the most part extremely well supported by the elegantly designed and executed experiments; TVM are unique in their mitochondrial function and SRC; memory T cells derived from an infection fail to display significant SRC and are instead more glycolytic like lytic effectors; this elevated SRC is not driven by FAO but rather is mediated downstream of IL-15 signaling; this SRC appears to be useful not for effector functions but for maintaining survival; SRC in TVM and Tmem actually increases during aging, dependent on IL-15.

In my view there are enough paradigm busting conclusions in this manuscript to keep us all talking for quite a while. One element that I appreciate about this manuscript is its concise simplicity, and as such, I think it is ready to go to press with the data it has (not sure I have ever written that before!)

that said, i have a couple of places where i think the authors might want to be more conservative in their conclusions as well as a couple of suggestions for experiments which, if done, would make their conclusions even stronger.

1) the authors go to some length to conclude that the best correlate with the SRC of TVMs is complex IV and not mitochondrial morphology or volume. however, the data show that TVM do have more mitochondria and that the elevation in SRC after aging does seem to correlate with a significant increase in fused mitochondria. in the absence of doing some sort of regression analysis between SRC and various parameters, i do not see how they have the data to say that the TVM SRC best tracks with the complex IV. however, it is not clear to me that there is any real need to... elevated SRC tracks with a number of features of mitochondria... number, health, and function, but NOT with FAO. this is the major point anyway, and there does not seem to me any real need to land on one dominant aspect of the mitochondria. to some degree they contradict themselves anyway, where, after describing the RNAseq data, they state that "these data indicate a shift in mitochondrial degradation versus generation in aging... resulting in increased mitochondrial load"; so is it load of complex IV? in my view this is neither here nor there, so tempering this conclusion will better fit the data in hand.

2) in the section on memory cells from virus infection, they critique a previous publication which showed that SRC partitioned into the TCM pool by stating that in that study, "TCM cells were defined as CD44^{hi}CD62L^{hi} CD8⁺ T cells obtained from mice after acute lymphocytic choriomeningitis virus (LCMV) infection, which would include TVM cells". in my re-reading of that data, they were using a P14 adoptive transfer system in which they assessed the metabolic function of only the transferred T cells. thus, after LCMV challenge, all should have become antigen experienced and few to no TVMs should have been present. point being, it does not look to me like the results they got could have been contaminated by TVMs and thus i am not sure that these cited data support the point they are trying to make. there may be others that do support this claim, and even if not, the authors do well to emphasize that any sorting scheme that does not make use of CD49d in its delineation of memory cells will inevitably capture TVMs and their attendant transcriptional /metabolic profile.

3) the authors conclude that "TMEM (IAV) cells that were used in Mito Stress assays in Figures 1 and 2 did not exhibit elevated SRC, despite the presence of TCM cells. Collectively, this dataset comprehensively demonstrates that high SRC is not a defining feature of antigen-experienced TMEM cells, regardless of whether they exhibit a TCM or TEM phenotype, but is instead a key characteristic of TVM cells." to be fair, TCM are only 10% of the total Tmem pool and thus, were they to have higher SRC, that would most likely be undetectable within the TEM metabolic phenotype by sheer numbers. the authors could support their conclusions further by actually sorting out Tcm and TEM from the IAV specific pool and then see what the SRC is like, but in the absence of this data, they should probably temper their conclusions on what metabolic phenotype antigen-specific TCM do and don't have.

this last point connects to my major suggestion which is to do the sorting on antigen specific TEM and TCM and actually get the answer on their metabolic profiles. as the data stand, the response of those that created the "Tmem have high SRC driven by FAO" are going to cry foul that the specific subset was not sorted out and verified. seems like a worthwhile point to be able to make.

second, if the TVM are not getting their energy for their SRC from FAO, then it would be worth knowing whether this is derived from glycolysis. poisoning glycolysis in various ways (2DG, 3 bromo pyruvate, etc) either in vivo before isolation of the cells or in vitro might be mechanistically informative on the metabolism side of things and add some novelty.

Response to Reviewers' comments

We thank the Reviewers for the positive comments as well as their insightful questions and comments, which we believe have substantially elevated the quality of this manuscript.

Reviewer 1

Remarks to the Author:

Quinn et al. are following previous work from their own lab and other groups in the determination of the metabolic signature of T cell subsets. They elegantly show the need for further analysis of T cell subsets to define the corresponding metabolic status and flexibility of subsets such as Naive, Virtual memory and MEM CD8⁺ T cells in mice. The work is sound and brings novel to another level that was incomplete from previous studies. This includes the link between SRC, cell longevity and the role of IL-15 and Bcl2 in this process.

#1: While the authors are correct that inclusion of further markers (such as CD49d) is important it is also important they discuss their own limitation. For instance, the study of ageing T cell in mice in relation to naive T cell as a precursor to other memory cells should take into account mice are not susceptible to thymic involution and that in humans, a pool of recent thymic emigrant T cells (RTE) are different metabolically from the pool of naive T cell with no antigenic-experience but with homeostatic proliferation experience. For instance, the interpretation "This study used individuals up to the age of 85 yo, which is significantly older than the human CD8⁺ T cells we have previously characterised (60-70 yo)¹⁷ and likely older than the equivalent mouse age of 15-18 mo characterised here." is flawed.

The reviewer has suggested that thymic involution does not occur in mice and that unactivated RTEs are metabolically distinct from unactivated mature naïve T cells. They are concerned that these two factors combined mean that it is flawed to use this data from ageing mice to extrapolate into ageing humans.

We highlight that thymic involution, which is defined as a loss of thymic tissue and a decrease in thymic output, is well documented in mice. Importantly, it occurs at around the same developmental age as in humans- in mice it begins at around 1 mo of age and continues at pace to ~3 mo, while in humans it begins in the first year of life and continues to middle age (35-45 yo; ~3%/yr) (Lynch, H.E. et al, *Trends Immunol*, 2009). Indeed, a direct comparison of the rates of thymic involution and thymic output, through the measurement of TCR delta excision circles (TRECs) has demonstrated analogous thymic involution in mice and humans (Sempowski et al, *Mol Immunol*, 2002; Sempowski et al, *J Immunol*, 2000; Gruver et al, *J Pathol*, 2007). We do note that it has been suggested that thymic involution may be more complete in humans as compared to mice and homeostatic proliferation is more evident at extreme ages (Yanes et al, *Semin Hematol*, 2017). In this model, the peripheral pool in aged humans is maintained *via* homeostatic proliferation of existing naïve cells, while there may be some residual thymic output in aged mice. Regardless, mice and humans are both seen to undergo a dramatic decline in RTE proportions with increasing age, which makes mice a reasonable model for human immune ageing in our study. Moreover, while metabolic differences between RTEs and mature naive CD8⁺ T cells have been noted after activation (Cunningham CA et al, *J Immunol*, 2018), the same study suggested that resting RTEs and mature T cells have similar metabolic profiles.

Finally, in addition to previously showing functional similarity between mouse and human CD8⁺ T cell subsets with age (Quinn et al, *Cell Rep*, 2018), we have now included data showing that phenotypic and metabolic patterns observed in mice are also observed in human cells. We

observed a significant increase in IL-15R β expression on CD8⁺ T cell subsets and a general increase in SRC with CD8⁺ T cells with age (albeit not statistically significant) (Fig. 6). This further suggests that the aging process in mouse and human CD8⁺ T cells is highly analogous. Collectively, while we acknowledge that our possible explanation provided for the apparent discrepancy between the work of Moskowitz et al. and our own data may not be correct, and we have tempered some of this language to stress this point, it is based on evidence and it is a reasonable hypothesis to propose and discuss.

#2: Also, the study dismissed the population of T Stem Cell Memory, which was shown to have the higher proliferative capacity among all T cells.

We agree that it would be highly informative to assess the metabolic profile of T_{SCM} cells and to correlate their SRC with proliferative capacity. Unfortunately, given the extremely low frequency of T_{SCM} cells *in vivo* and the large numbers of cells required for assaying metabolic activity using the XFe96 Seahorse platform, a direct assessment of SRC in T_{SCM} cells is not feasible at present. We have added a sentence in the discussion noting the relevance of this analysis.

“To test this paradigm further, it would be highly informative to assess SRC in other T cell subsets, such as stem cell memory T cells, which are highly proliferative, and resident memory T cells, which retain a level of proliferative capacity *in situ* and are highly IL-15 dependent. Unfortunately, the low frequency of these cells precludes their analysis using the current Seahorse technology.”

#3: Expression of IL-15R subunits should be measured in all subsets and in the context of aging, in human T cells to at least partially validate the mechanisms in human.

We have now performed this analysis and included it as Figure 6a,b. Similar to observations in mice, we also found that human T_{VM} cells expressed the highest levels of IL-15R β and that this expression was broadly increased with age. We include the following section in the Results:-

“Elevated IL-15R β Expression and SRC in Elderly Human CD8⁺ T cells

To determine whether our findings in mice were relevant to humans, we first analysed expression of IL-15R β on young adult (20-30 yo) and older adult (60-80 yo) human CD8⁺ T cells. We found significantly higher expression of IL-15R β on T_{VM} cells compared to T_N cells in young adults, and this expression increased significantly with age across all subsets (Fig. 6a,b), similar to our observations in mice. Moreover, when total CD8⁺ T cells from young adults and older adults were assayed for SRC, we found a trend toward increased SRC with advanced age (p=0.1) (Fig. 6c,d). Given that we have previously described a loss of proliferative capacity in CD8⁺ T cells from older humans³⁵, these data suggest a similar lack of correlation between SRC and functionality in human CD8⁺ T cells as observed for mouse cells. Moreover, these data indicate a correlation between elevated SRC in human CD8⁺ T cells and an age-related increase in IL-15 sensitivity.”

#4: The study should measure the expression of co-stimulatory molecules such as Cd28 and Cd27 as well as other well-accepted markers of T cell differentiation. For instance, Cd28 was shown to lead to Akt activation which in turn is involved in glycolysis. This is important in the context of the metabolic hypothesis.

While downregulation of CD28 and CD27 are key markers of terminally differentiated T cells in humans, it is not clear that this combination of markers functions in the same way in mouse cells and consequently we do not routinely assess them.

With regard to other markers of T cell differentiation, our previous publication (Quinn et al, 2018, Cell Rep) has performed transcriptional analysis and phenotypic characterisation of young and aged CD8⁺ T cell subsets, with a particular focus on the expression of co-stimulatory molecules in the context of investigating an 'exhausted' phenotype. We do not see substantial shifts, in either direction, of co-stimulatory molecule transcripts (PD1, CTLA4, LAG3, TIM3, ICOS, CD28) on T_{VM} cells with age.

We acknowledge that expression of CD28 has indeed been linked to augmented glycolytic capacity through Akt-dependent and independent pathways in activated T cells. However, this study is primarily focused on oxidative phosphorylation capacity in resting T cells. We are not aware of literature suggesting that expression of CD28 impacts respiratory capacity of resting T cells, as distinct from activated T cells, so we regard analyses of co-stimulatory markers and glycolysis on resting T cell subsets outside the scope of the current study.

#5: This study focusses on the phenomenon of mitochondrial fusion but dismissed the fission. Both should be included and discussed.

We have focused on the fused mitochondrial state primarily because fused mitochondria have previously been associated with the T_{MEM} phenotype and an increased SRC, both of which were being assessed here. Of course, we agree that both processes are relevant to the overall outcome of increased mitochondrial capacity. To better reflect this, we have revised our statements in Results and Discussion as follows:-

Results: “Analysis of transcript levels associated with mitochondrial fusion (*Mfn1*, *Mfn2*, *Opa1*) or fission (*Dnm1l*) revealed minimal to no change across T cell subsets or with aging (Supplementary Fig. 1C). Collectively, whilst true delineation of the impact of mitochondrial dynamics on mitochondrial load requires more detailed biochemical analyses, these transcriptional data highlight that there may be a decrease in mitochondrial degradation and an increase in biogenesis with age, particularly in the T_{VM} subset, which may drive the observed increase in mitochondrial load and SRC.”

Discussion: “In the T cell field, mitochondrial fusion has been largely associated with advantageous outcomes; namely elevated SRC and memory T cell formation^{3,6}, although this association was not observed in this study, while a more fragmented (or fissioned) mitochondria is associated with effector T cells⁶. However, while fission of mitochondria is essential for cytokinesis and for isolating dysfunctional mitochondria to allow autophagic degradation⁵⁴, fusion of mitochondria is more generally considered....”

Reviewer 2

Remarks to the Author:

This study by Quinn et al. aims to address the notion that increased SRC is intimately linked to increased CD8 T cell function, a suggestion made, like much of the widely accepted immunometabolism data, from in vitro generated cells and confounded population identifications. This is a very important and well performed body of work that, primarily, re-attributes increased SRC among CD8 T cells to the virtual memory (rather than traditional Ag stimulated memory). This work uses direct ex vivo analysis and not only delineates and breaks

apart a widely accepted generalizations regarding memory T cells and SRC, mitochondrial morphology and fatty acid oxidation, but provides a starting point from which to reassess what has been proposed in immunometabolism. It has been long known that memory CD8 T cells generated in different infection systems can result in distinct outcomes; why the field has accepted generalizations regarding metabolism from in vitro generated effector and memory T cells remains unclear, and this paper aids us in calling those findings into question. This is a foundation from which we can truly approach the nuanced field of T cell memory with an interest in metabolism.

#6: That said, the authors correlate IL-15 dependence and increased SRC, while this is the case for their population of virtual memory cells, I would argue that the authors have demonstrated that IL-15 is necessary for SRC in virtual memory cells (an IL-15 dependent population), however they have not demonstrated that IL-15 is sufficient for SRC increase. In the interest of avoiding new mis-generalizations in the field, I would encourage the authors to either, recalibrate the language to be very clear about what is and is not shown but correlated or evaluate other populations of IL-15 dependent cells for SRC.

We agree that this distinction is a key point. We have recalibrated our language to include the following statement:-

“Additionally, while our results show that IL-15 is necessary for high SRC in T_{VM} cells, they do not show that it is entirely sufficient, and it is possible that there may be additional signalling pathways that co-ordinate with IL-15 to drive increased SRC.”

#7: The latter would prove highly difficult, though Thomas Kupper has demonstrated feasibility. For example, TRM cells from various organs after LCMV have been demonstrated to be IL-15 dependent or independent (Schenkel & Fraser et al. 2016), do these distinctions correlate with increased SRC?

We wholeheartedly agree that it would be interesting to assess the population of T_{RM} cells that have been shown to be IL-15 dependent. Unfortunately, to directly assess SRC or mitochondrial characteristics that best correlate with SRC, we need large numbers of cells (in our current protocols, we use at least 1 x 10⁶ cells), which would not be feasible with current T_{RM} cell-inducing infection models. To acknowledge this limitation in our dataset, we have noted the following:

“To test this paradigm further, it would be highly informative to assess SRC in other T cell subsets, such as stem cell memory T cells, which are highly proliferative, and resident memory T cells, which retain a level of proliferative capacity *in situ* and are highly IL-15 dependent. Unfortunately, the low frequency of these cells precludes their analysis using the current Seahorse technology.”

Finally, the breakdown of mitochondrial morphology holding some bearing on function and etomoxir dosing are very important points of this paper.

#8: Major Concern: Mentioned above regarding IL-15 and SRC being correlated in a IL-15 dependent population as required for SRC, however not necessarily sufficient for SRC increase. Explore further or adapt language accordingly.

We have revised this as described above in point **#6**.

#9: Major focus on spare respiratory capacity, however it is unclear, is this a physiologically relevant function or just a correlate? Please clarify within the text.

The reviewer raises an interesting and unresolved point in T cell biology. It remains unclear whether SRC in resting T cells is physiologically relevant to their subsequent cell function - such as proliferation, cytokine production and cytotoxicity - after stimulation.

We have focused on SRC as others have contended that this metabolic parameter i) can define T_{MEM} cells as distinct from antigenically naïve and effector cell subsets, and ii) is predictive of CD8⁺ T cell function (van der Windt et al, *PNAS*, 2013, Buck et al, *Cell*, 2016). In contrast, our analyses demonstrate that it serves neither of these purposes - T_{VM} cells are antigenically naïve and yet have the highest SRC; T_{MEM} subsets were likely conflated with T_{VM} cells, leading to overestimation of their SRC; T_{VM} cells increase in SRC but lose functionality with age, highlighting that SRC and T cell functionality are not coupled.

While our data ostensibly suggests that SRC is poorly associated with T cell functional capacity, it is also possible that SRC does in fact reflect the ‘theoretical’ capacity of the cell to respond to stimuli, which is not realized due to confounding factors that prevent the utilization of that capacity. We also believe that there is better evidence to indicate that SRC is linked to cell longevity rather than functionality *per se*. At this point, we conclude that SRC is not a correlate of functionality but remains a correlate of longevity, and further studies are needed to determine whether this correlation is established through causation.

We have included the following clarifying statements in the Introduction and the Discussion:-

Introduction: “SRC is the difference between basal and maximal oxygen consumption rates (OCR)^{4,5} and it reflects the mitochondrial capacity that a cell holds in reserve, which may mitigate stress from sudden increases in energy demand. Increased SRC has been proposed to mediate both i) enhanced T cell functionality, in the form of metabolic memory that confers immediate responsiveness after secondary antigen exposure⁵, and ii) the increased longevity of T_{MEM} cells^{5,6}.”

Discussion: “Of note, this data highlights that high SRC is not necessarily linked to superior CD8⁺ T cell function. Previously, high SRC was proposed to facilitate accelerated proliferation and increased cytokine production with *in vitro* generated T_{MEM} cells⁵. We and others have previously demonstrated that proliferative capacity is highest in T_{VM} cells, followed by T_N and T_{MEM} cells directly *ex vivo* in young individuals^{9,17,19,22}, which tracks with their SRC. However, any correlation breaks down with ageing, as proliferative capacity is highest in T_N cells and relatively poor in T_{VM} and T_{MEM} cells^{17,22} and the latter two populations exhibit elevated SRC. We propose that that this lack of correlation between SRC and function in aging is compounded by an age-related defect in TCR signalling that uncouples metabolic potential in resting cells from metabolic engagement and subsequent functionality in cells after activation. We and others have shown that there is specific inhibition of TCR-driven proliferation in aged T_{VM} cells and dysregulation of signalling cascades downstream of TCR engagement with age^{17,22,44,45}. This dysregulation may prevent aged T_{VM} cells from efficiently engaging their increased mitochondrial reserve upon TCR signalling to facilitate more rapid proliferation..... There is perhaps better evidence that SRC might be a physiologically relevant indicator of T cell longevity, rather than functionality after activation. The elevated

SRC observed in T_{VM} cells, which increases with age, is paralleled by our previous data suggesting these cells have a survival advantage both *in vivo* and *in vitro*^{13,17}. Moreover, IL-15 is both necessary for elevated SRC and essential for T_{VM} cell survival¹⁵. Finally, SRC has been found to be directly responsible for survival of myocytes under conditions of hypoxia or nutrient deprivation⁴⁶. Despite this, further studies are required to determine whether increased SRC is responsible for enhanced T cell survival.”

#10 The antigen depot known to be associated with IAV makes the day 20 timepoint of ‘memory’ cells in figure 2B difficult to interpret. I would recommend either using LCMV infected cells and memory at a d30 time point or later date after IAV. The increase in glycolysis but not OCR indicates this time point may simply be too early and these are indeed confounded with an effector population.

The analysis of ‘early memory’ at d20 was performed to (i) enable the detection of a sufficient number of true memory cells (on the basis of evidence that IAV-derived pMHC1 presentation has completely ceased by at least d11-14 after infection (Mintern J et al, *J Immunol*, 2009)) and (ii) be able to detect any medium term effects of infection/inflammation that may have been lost in very long term memory, such as the increased SRC in T_{VM} cells.

However, we also acknowledge that this is a short-term memory timepoint. Therefore, in our new, more specific analyses of T_{CM} and T_{EM} cells, we have harvested these memory populations from mice infected with LCMV at d60 after infection. As observed at d20 after IAV infection, we found that T_{CM} and T_{EM} cells harvested at day 60 after LCMV infection still had a significantly lower SRC than T_{VM} cells and we have included the results as Fig. 3e,f.

Minor Concerns:

#11 Please state in the text or legend where TMEM cells are derived from in Figure 1 & 2. It is stated but not until line 180 at a transitional point. Reader should not have to search different sections for this information.

We apologize for this oversight. We have included the following in the legend to Figures 1 and 2, “...isolated from the spleens of naïve young and aged SPF mice.”

#12 Authors state that the SRC tracked closely with CIV. This does not appear to be the case in a generalizable way. (Fig 1g & 1a-b) Specifically, TN cells demonstrated an increase in CIV between young and aged mice but have no increase in SRC.

We acknowledge that too great an emphasis was placed on this point and that this is neither an overwhelming correlation, nor a critical finding of the study. Therefore, we have removed the data showing supercomplexes with CI and have tempered the language around this to the following:-

“...and, although not an absolute correlation, the amount of CIV appeared to correlate better with SRC than mitochondrial load or morphology; namely CIV was increased in young T_{VM} compared to T_N cells, and age-related increases in CIV were most marked in the T_{VM} population (Fig. 1g). Collectively, our analyses of mitochondrial load and morphology suggested that they were broadly predictive of SRC, in particular being associated with age-related increases in SRC. Expression levels of ETC CIV appeared to most accurately predict cellular SRC across age and subsets.”

#13 Line 76, CD44 should not be described as a marker of memory, necessarily, because it comes up early during activation as well.

This has been changed to "...high levels of CD44, a classical marker of activation, but low levels of CD49d....."

#14 Figure 1B would be clearer to call the y-axis, SRC (OCRmax-OCRbasal).

This has been revised throughout.

Reviewer #3 (Remarks to the Author):

This is an exceptionally written, experimentally concise and compelling manuscript detailing the metabolic profiles of naive and memory T cell subsets, in young and aged mice. In the process, it details some very important findings regarding the role of mitochondrial metabolism and FAO in the regulation of antigen-experience and -inexperienced T cell memory survival. The data they present continues to shake up the existing paradigm that is based almost exclusively on highly contrived in vitro models and inappropriate usage of metabolic inhibitors to promote the model that memory T cells have elevated SRC as a result of elevated FAO. The present manuscript punches holes in both of these assertions, and in a fashion that cannot be easily disregarded. Collectively, the conclusions of the manuscript are for the most part extremely well supported by the elegantly designed and executed experiments; TVM are unique in their mitochondrial function and SRC; memory T cells derived from an infection fail to display significant SRC and are instead more glycolytic like lytic effectors; this elevated SRC is not driven by FAO but rather is mediated downstream of IL-15 signaling; this SRC appears to be useful not for effector functions but for maintaining survival; SRC in TVM and Tmem actually increases during aging, dependent on IL-15.

In my view there are enough paradigm busting conclusions in this manuscript to keep us all talking for quite a while. One element that I appreciate about this manuscript is its concise simplicity, and as such, I think it is ready to go to press with the data it has (not sure I have ever written that before!)

That said, I have a couple of places where I think the authors might want to be more conservative in their conclusions as well as a couple of suggestions for experiments which, if done, would make their conclusions even stronger.

#15 The authors go to some length to conclude that the best correlate with the SRC of TVMs is complex IV and not mitochondrial morphology or volume. However, the data show that TVM do have more mitochondria and that the elevation in SRC after aging does seem to correlate with a significant increase in fused mitochondria. In the absence of doing some sort of regression analysis between SRC and various parameters, I do not see how they have the data to say that the TVM SRC best tracks with the complex IV. However, it is not clear to me that there is any real need to... elevated SRC tracks with a number of features of mitochondria...

number, health, and function, but NOT with FAO. this is the major point anyway, and there does not seem to me any real need to land on one dominant aspect of the mitochondria. to some degree they contradict themselves anyway, where, after describing the RNAseq data, they state that "these data indicate a shift in mitochondrial degradation versus generation in aging... resulting in increased mitochondrial load"; so is it load of complex IV? in my view this is neither here nor there, so tempering this conclusion will better fit the data in hand.

We acknowledge that we placed too great an emphasis on this point that is neither an overwhelming correlation nor a critical finding of the study. Therefore, we have removed the data showing supercomplexes with CI and have tempered the language around this (below). However, given that a main driver for analysing mitochondrial characteristics was to determine their association with SRC, we did wish to offer some interpretation of the results in light of how they tracked with SRC, and so have modified our statements as follows:-

"...and, although not an absolute correlation, the amount of CIV appeared to correlate better with SRC than mitochondrial load or morphology; namely CIV was increased in young T_{VM} compared to T_N cells, and age-related increases in CIV were most marked in the T_{VM} population (Fig. 1g). Collectively, our analyses of mitochondrial load and morphology suggested that they were broadly predictive of SRC, in particular being associated with age-related increases in SRC. Expression levels of ETC CIV appeared to most accurately predict cellular SRC across age and subsets."

#16 In the section on memory cells from virus infection, they critique a previous publication which showed that SRC partitioned into the TCM pool by stating that in that study, "TCM cells were defined as CD44^{hi}CD62L^{hi} CD8⁺ T cells obtained from mice after acute lymphocytic choriomeningitis virus (LCMV) infection, which would include TVM cells". in my re-reading of that data, they were using a P14 adoptive transfer system in which they assessed the metabolic function of only the transferred T cells. thus, after LCMV challenge, all should have become antigen experienced and few to no TVMs should have been present. point being, it does not look to me like the results they got could have been contaminated by TVMs and thus i am not sure that these cited data support the point they are trying to make. there may be others that do support this claim, and even if not, the authors do well to emphasize that any sorting scheme that does not make use of CD49d in its delineation of memory cells will inevitably capture TVMs and their attendant transcriptional /metabolic profile.

While we acknowledge that it is somewhat difficult to interpret from the paper as written, we believe that the specific experiment in Phan et al to which we refer (Figure 6D) describes endogenous CD8⁺ T cell populations. There is no mention of TCR transgenic (P14 cells) in the legend or text, nor is there any suggestion that cells were adoptively transferred, which would then imply the use of P14 cells. The experiment is described and interpreted as follows:-

"...we sort purified WT Tcm and Tem cells (CD44^{hi}CD62L^{hi} and CD44^{hi}CD62L^{lo}, respectively) from mice previously infected with LCMV at least 60 days prior and measured ECAR and OCR via extracellular flux analysis (Figure 6D). WT Tem cells maintained significantly lower SRC and a reduced basal rate of cellular respiration compared to WT Tcm cells."

However, we also acknowledge that the experiment described in Figure 6D may have been performed following P14 cell transfer and that the manuscript has not described it as such. Therefore, we have revised our description to allow for this ambiguity as follows:-

“In that study, T_{CM} cells appear to have been defined as $CD44^{hi}CD62L^{hi} CD8^{+}$ T cells obtained from mice after acute lymphocytic choriomeningitis virus (LCMV) infection, which would include T_{VM} cells²⁰.”

“This highlights the possibility that $CD8^{+}$ T cell populations previously defined as T_{CM} cells, from young, aged or infected mice, may have been predominantly comprised of T_{VM} cells.”

“Subsequent work proposed that high SRC partitioned preferentially with the T_{CM} cell population but T_{CM} cells in this study also appeared to be identified as $CD8^{+}CD44^{hi}CD62L^{hi}$ cells from LCMV infected mice⁷.”

#17 The authors conclude that "TMEM (IAV) cells that were used in Mito Stress assays in Figures 1 and 2 did not exhibit elevated SRC, despite the presence of TCM cells. Collectively, this dataset comprehensively demonstrates that high SRC is not a defining feature of antigen-experienced TMEM cells, regardless of whether they exhibit a TCM or TEM phenotype, but is instead a key characteristic of TVM cells." to be fair, TCM are only 10% of the total Tmem pool and thus, were they to have higher SRC, that would most likely be undetectable within the TEM metabolic phenotype by sheer numbers. the authors could support their conclusions further by actually sorting out Tcm and TEM from the IAV specific pool and then see what the SRC is like, but in the absence of this data, they should probably temper their conclusions on what metabolic phenotype antigen-specific TCM do and don't have.

this last point connects to my major suggestion which is to do the sorting on antigen specific TEM and TCM and actually get the answer on their metabolic profiles. as the data stand, the response of those that created the "Tmem have high SRC driven by FAO" are going to cry foul that the specific subset was not sorted out and verified. seems like a worthwhile point to be able to make.

We agree with the reviewer that this is the ideal experiment to support our contention of a lack of association between SRC and conventional memory phenotype (either T_{EM} or T_{CM}). We have now included this experiment as Fig. 3e,f. Young SPF mice were either left uninfected or infected with LCMV and, at d60, the metabolic profiles of T_N and T_{VM} cells from naïve mice and T_{CM} and T_{EM} cells from LCMV-infected mice were analysed. The results from this analysis verified our earlier findings that T_{VM} cells exhibit the highest SRC, and yielded a more precise analysis of the distinction between T_{CM} and T_{EM} cells. The following passage has been included in the Results:-

“To definitively determine the relative SRC of T_{CM} and T_{EM} cells alongside T_{VM} cells, we performed the Seahorse MitoStress assay on T_N and T_{VM} cells isolated from naïve SPF mice and T_{EM} and T_{CM} cells isolated from LCMV-infected mice at 60 days post infection. As previously observed, T_{VM} cells had a significantly higher SRC than all other subsets, T_{EM} cells had a significantly lower SRC than all other subsets, and T_{CM} cells had a modestly higher SRC than T_N cells (Fig. 3e, f). Collectively, this dataset comprehensively demonstrates that high SRC is not a defining feature of conventional T_{MEM} cells, in particular the T_{EM} cell subset, but is instead a key characteristic of T_{VM} cells.”

#18 Second, if the TVM are not getting their energy for their SRC from FAO, then it would be worth knowing whether this is derived from glycolysis. poisoning glycolysis in various ways (2DG, 3 bromo pyruvate, etc) either in vivo before isolation of the cells or in vitro might be mechanistically informative on the metabolism side of things and add some novelty.

Given ours and others' observations that the high SRC in T cells was not being fuelled by FAO, we performed metabolic profiling of T_{VM} cells in the presence of 2-DG to determine whether a substrate generated by glycolysis was driving the elevated SRC. While we did note a reduction in extracellular acidification rate (ECAR) with 2-DG, verifying its activity, we did not observe a notable reduction in OCR, suggesting that products of glycolysis were not required to drive the elevated SRC. This data has been included as Supp Fig 2 along with a description of the data in the Results as follows:-

“We next investigated the possibility that glycolysis was required to fuel the high SRC observed in T_{VM} cells, most likely via the production of pyruvate²⁹. The MitoStress assay was performed on T_{VM} cells from naïve mice with the addition of 2-Deoxy-D-glucose (2-DG), a glucose analog that inhibits glycolysis. The addition of 2-DG had a minimal effect on OCR (Supplementary Fig. 2a), similar to the addition of low dose etomoxir (Fig. 1h). By contrast, 2-DG addition resulted in a dramatic reduction in ECAR, which confirms its effective inhibition of glycolysis (Supplementary Fig. 2b). Together, these data suggest that the high basal SRC observed in T_{VM} cells is not exclusively dependent on either FAO or glycolysis but may be fuelled by a substrate generated independently of both pathways.”

REVIEWERS' COMMENTS:

Reviewer #2 (Remarks to the Author):

i am satisfied

Reviewer #3 (Remarks to the Author):

the previous issues have been resolved to my satisfaction and i congratulate the authors on their responsiveness to the reviews and a very nice manuscript revision. I think this is going to get the kind of attention it deserves.